# Cognitive effort investment: Does disposition become action?

**Corinna Kührt**[1] *, **Sven-Thomas Graupner**[1], **Philipp C. Paulus**[2], **Alexander Strobel**[1]

**1** Faculty of Psychology, Technische Universität Dresden, Dresden, Germany, **2** Department of Psychology, University of Freiburg, Freiburg, Germany

* corinna.kuehrt@tu-dresden.de

## Abstract

Contrary to the law of less work, individuals with high levels of need for cognition and self-control tend to choose harder tasks more often. While both traits can be integrated into a core construct of dispositional *cognitive effort investment*, its relation to actual cognitive effort investment remains unclear. As individuals with high levels of cognitive effort investment are characterized by a high intrinsic motivation towards effortful cognition, they would be less likely to increase their effort based on expected payoff, but rather based on increasing demand. In the present study, we measured actual effort investment on multiple dimensions, i.e., subjective load, reaction time, accuracy, early and late frontal midline theta power, N2 and P3 amplitude, and pupil dilation. In a sample of $N = 148$ participants, we examined the relationship of dispositional cognitive effort investment and effort indices during a flanker and an *n*-back task with varying demand and payoff. Exploratorily, we examined this relationship for the two subdimensions *cognitive motivation* and *effortful-self-control* as well. In both tasks, effort indices were sensitive to demand and partly to payoff. The analyses revealed a main effect of cognitive effort investment for accuracy (*n*-back task), interaction effects with payoff for reaction time (*n*-back and flanker task) and P3 amplitude (*n*-back task) and demand for early frontal midline theta power (flanker task). Taken together, our results partly support the notion that individuals with high levels of cognitive effort investment exert effort more efficiently. Moreover, the notion that these individuals exert effort regardless of payoff is partly supported, too. This may further our understanding of the conditions under which person-situation interactions occur, i.e. the conditions under which situations determine effort investment in goal-directed behavior more than personality, and vice versa.

## Introduction

In general, people prefer a less demanding behavioral option when it is rewarded as much as a more demanding one–but not all. Hull [1] referred to the general tendency of avoiding or at least reducing energy expenditure or work as the *law of less work*. Yet, this principle of behavior is not only limited to physical work, but is true for cognitive work as well. Work–or effort–even reduces the value associated with a given payoff–a phenomenon called *effort discounting*

**Editor:** Árpád Csathó, University of Pecs Medical School, HUNGARY

**Data Availability Statement:** All relevant files are available from the OSF repository (https://osf.io/mw4jd/).

**Funding:** This research was funded by the German Research Foundation (Deutsche

Forschungsgemeinschaft, DFG; SFB 940/2 project B6). The funders had no role in study design, data collection and analysis, decision to publish, or preparation of the manuscript.

**Competing interests:** The authors have declared that no competing interests exist.

[2]. While numerous studies confirm this phenomenon as a basic principle [e.g., 3], individual differences exist: In a study by Kool et al. [4], trait *self-control* (i.e., the capacity of adapting one's immediate state to achieve higher-order goals [5]) was negatively related to the extent of demand avoidance in a demand selection task. In another study by Westbrook et al. [6], trait *need for cognition* (i.e., the willingness to invest cognitive effort and enjoy it [7]) was negatively related to effort discounting in a cognitive effort discounting paradigm. Consequently, both self-control and need for cognition may play an important role in explaining individual differences in cognitive effort investment.

Self-control and need for cognition are moderately correlated, $r$ = .28, but share the common core of goal directedness [8]. In an effort to capture this common core, Kührt et al. [8] proposed a hierarchical model that integrates self-control and need for cognition in a core construct of *cognitive effort investment*. Fig 1 illustrates this model: two first-order factors, i.e., *cognitive motivation* as indicated by need for cognition and intellect (i.e., the motivation to intellectual achievements [9]), and *effortful self-control* as indicated by self-control and effortful control (i.e., attention control [10]) give rise to a second-order factor of cognitive effort investment. Kührt et al. [8] defined dispositional cognitive effort investment as "self-reported dispositional differences in the willingness and tendency to exert effortful control". This core construct model provides an integrative and pragmatic assessment of individual differences in cognitive effort investment. However, its relation to behavior in terms of the amount of cognitive effort actually invested (henceforth actual cognitive effort investment) remains unclear.

Do individuals with high cognitive effort investment actually invest effort more efficiently (i.e., dynamically according to task demands) in task processing and regardless of external incentives (e.g. payoff)? By definition these individuals are highly intrinsically motivated to perform effortful cognition [8], approach and enjoy cognitive challenges while reporting less

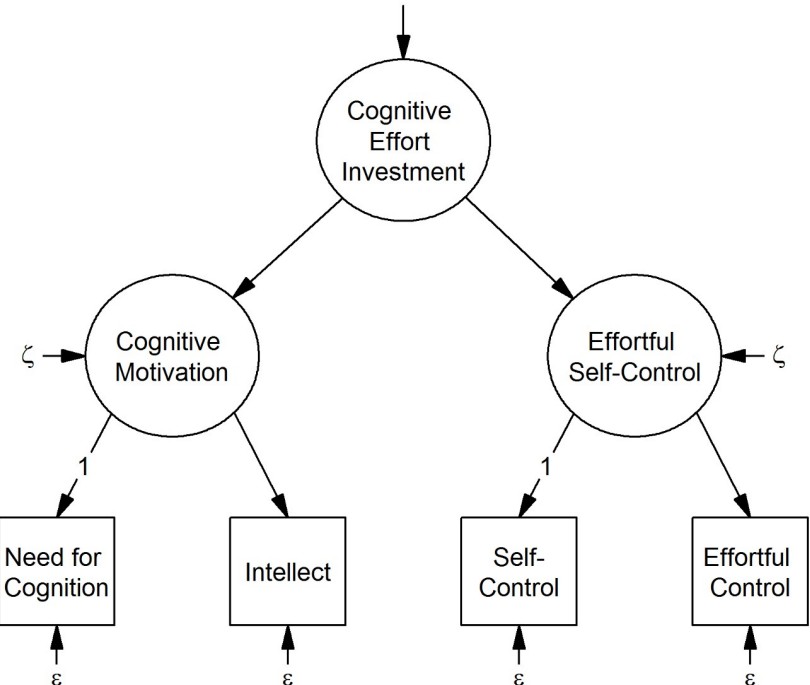

**Fig 1. Core construct of cognitive effort investment.** ξ residuals of the first-order factors set equal, ε error variances of the indicator variables set equal, 1 loadings of the indicator variables fixed to 1.

difficulty and less negative emotions [11] and perform self-controlled behavior relatively effortlessly and unconsciously [12]. In line, Westbrook et al. [6] experimentally showed that individuals with high need for cognition (i.e., a main indicator of cognitive effort investment) indicated a higher subjective value with demanding tasks than individuals with low need for cognition. However, individuals with high need for cognition not only value demanding tasks more than those with low need for cognition, but also process these tasks with less effort as demonstrated by Mussel et al. [13]. They found that individuals with high levels of need for cognition, as a main component of cognitive effort investment, allocated their cognitive resources according to task demands, as indicated by frontal midline theta power (FMθ) in the electroencephalogram, while performing better in the harder task. This was not the case for individuals with low levels of need for cognition, who invested a considerable amount of resources even in easy tasks and performed comparably. In a study by Sandra and Otto [14], additional extrinsic motivation in terms of payoff did not add to the already high intrinsic motivation in individuals with high need for cognition to invest effort. They observed that individuals with low levels of need for cognition increased their amount of effort expenditure when facing higher payoff, whereas individuals with high need for cognition did not. According to the available evidence, individuals with high levels in cognitive effort investment would be less likely to increase their effort based on increasing payoff, but rather based on increasing demand. In other words, when given equal payoffs, individuals with high compared to low cognitive effort investment invest effort according to task demands; whereas when given equal demands, they exert less effort, especially under low demand conditions. Thus, their performance would not benefit from payoff in terms of enhanced resource allocation (e.g. stronger increase in FMθ in the electroencephalogram) in order to improve performance (e.g. faster reaction times). Fig 2 hypothetically illustrates these interactions.

One might assume that not cognitive effort investment–here need for cognition–, but rather elevated cognitive functioning may alternatively explain these findings. Perceptual speed, willingness to switch tasks, working memory capacity and crystallized intelligence may affect task processing, e.g. RTs, flexible adaptation to demand levels in the flanker task, identification of the target stimuli in the *n*-back task, etc. Indeed, traits related to cognitive effort investment, such as need for cognition, intellect and self-control, typically show medium to large correlations with fluid intelligence, crystallized intelligence and school grades [9, 15–17]. However, in inhibitory control tasks, such as the flanker task, need for cognition is not related to performance (Kendall's τ -.04 [18]). Accordingly, Mussel et al. [13] demonstrated that the evidence for trait need for cognition still remained after controlling for working memory capacity which did not moderate the relation between allocation of cognitive resources and

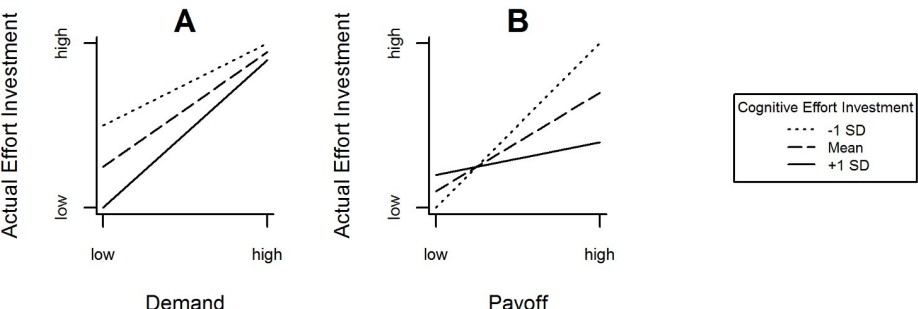

**Fig 2. Expected relation between actual effort investment and (A) demand and (B) payoff as a function of cognitive effort investment.**

demand. Additionally to need for cognition, Sandra and Otto [14] also analyzed the impact of cognitive capacity on the relation between cognitive effort investment and payoff. They found that individuals with low executive function invested more cognitive effort with increasing payoff, whereas individuals with high executive function did not. This observation on cognitive capacity contrasts with that on trait need for cognition and implies that the findings, reported on the differences in need for cognition, actually relate to cognitive motivation rather than capacity.

The aim of the present study is to further investigate the relationship between trait cognitive effort investment and actual effort investment during task processing. Consequently, here we investigate (1) whether cognitive effort investment is related to objective effort indices and (2) whether it moderates the relation between effort indices and demand or payoff, respectively, in typical cognitive control tasks. To this end, we apply an *n*-back and a flanker task. Both tasks allow the manipulation of different demand and payoff levels and were previously used in this context. Moreover, using two different executive functions, i.e. updating in the *n*-back task and inhibition in the flanker task, will provide us with a detailed and more generalizable picture of the interrelations. To measure effort investment during task performance, we combine self-report with objective measures of effort exertion including behavioral, electrocortical and psychophysiological indices. This multi-dimensional approach aims at providing a more complex picture of actual effort investment during task performance. In the following, we briefly explain the role of each indicator within the context of effort and how it is moderated by traits related to cognitive effort investment.

The easiest and most intuitive way to access effort experience is to ask participants directly how they perceive task load. The NASA Task Load Index (NASA-TLX) [19] reliably and validly accesses perceived load on six dimensions. Indeed, individuals report increasing load for greater demand levels in the *n*-back task [6, 20]. However, we could not find any paper that systematically examines differences in perceived load due to traits related to cognitive effort investment.

Reaction time and accuracy are typical behavioral indices of task performance, easy accessible and well-studied. Thus, both are ideal indicators of cognitive effort. Typically, reaction time increases and accuracy decreases with increasing load in the *n*-back task [6, 13], in the flanker task [21] and in a combined *n*-back task with flankers [22]. Kramer et al. [20] found a positive relation between need for cognition and performance (d') for all *n*-back levels (i.e. 1-back, 2-back, 3-back), $.14 < r < .22$, in adolescents. Furthermore, Mussel et al. [13] reported that the decrease in accuracy with increasing *n*-back level was weaker for individuals with high compared to low need for cognition. As to payoff, Sandra and Otto [14] examined switch costs (i.e., reaction time difference between task switches and repetitions) in a task-switching paradigm, whereas reduced switch costs indicate increased cognitive investment. Accordingly, higher incentives lead to a switch cost increase in individuals with high levels of need for cognition compared to a reduction in individuals with low levels [14].

Electroencephalography (EEG) allows for non-invasive insights into functional and temporal brain activity during task performance. A reliable marker of cognitive control is FMθ reflecting active processing in response to high cognitive demand [23, 24]. The N2 amplitude mirrors this spectral activity on the level of event-related potentials (ERP) [23, 25]. Another promising ERP component is the P3 indicating the allocation of mental resources [26–28].

As to FMθ, the frequency in EEG power ranges between approximately 4–7 Hz [24, 29]. Several studies [e.g., 22, 30] and reviews [23, 24, 29] suggest an increase in FMθ with increasing task demands in a variety of task, e.g. *n*-back, flanker, Go/NoGo, Simon. Mussel et al. [13] examined the relation between need for cognition, i.e. a main component of cognitive effort investment, and FMθ in an *n*-back task. The authors identified two temporally separable

processes in the theta band. In the early phase (0–650 ms) FMθ increases indicating evaluation and response preparation, whereas in the late phase (650–1900 ms) FMθ decreases indicating relaxation and preparation for the next trial. With increasing demand, early FMθ decreases and late FMθ increases. Regarding trait effects, the results revealed different patterns for high and low need for cognition: late FMθ in individuals with high levels of need for cognition was lower in the easiest condition compared to low levels and increased linearly with *n*-back level. The authors interpret these results as those individuals with high levels of need for cognition efficiently allocate cognitive resources according to task requirements in contrast to individuals with low levels.

In line with the FMθ literature, the N2 amplitude increases with increasing demand in different cognitive control tasks, e.g. Go/NoGo, flanker, stop signal, Stroop [25, 30]. Forster et al. [21] observed a linear relationship between N2 amplitude and demand in a flanker task. However, the two medium demand levels revealed no difference. So far, there is no literature that systematically investigated the effects of traits related to cognitive effort investment on this relationship.

While the N2 seems to indicate premotor cognitive processes, the P3 appears to reflect response-related, evaluative processes [31]. In a hybrid *n*-back/flanker task, Scharinger et al. [22] observed the P3 amplitude to decrease with increasing demand, i.e. both increasing *n*-back level and flanker incongruency. As to trait effects, the P3 amplitude is larger in individuals with high than low levels of need for cognition, i.e. a main component of cognitive effort investment, in a novelty oddball paradigm [32].

A prominent and reliable marker of cognitive effort exertion is pupil dilation [33, 34], which is additionally sensitive to payoff [35]. Pupil dilation increases with demand in a variety of tasks, e.g. *n*-back, flanker, Go/NoGo [22, 33]. Moreover, it increases with payoff in a cognitive control tasks [36]. Yet, the moderating role of traits related to cognitive effort investment have not been examined in this context.

Within the present paper, we investigate–as mentioned before–(1) whether cognitive effort investment is related to objective effort indices (i.e., subjective load, reaction time, accuracy, early and late FMθ power, N2 and P3 amplitude, and pupil dilation) and (2) whether it moderates the relation between effort indices and demand or payoff, respectively, in the *n*-back and flanker task even after controlling for cognitive abilities and crystallized intelligence. To the first question, we determine (H1) whether cognitive effort investment reliably predicts each effort index in the *n*-back and flanker task. Moreover, (H2) cognitive effort investment is negatively correlated with perceived task load regarding the cognitive task battery and the flanker task. To the second question, we examine (H3) whether each effort index shows a stronger change in individuals with high (vs. low) cognitive effort investment with increasing demand in the *n*-back and flanker task; and (H4) whether each effort index changes less in individuals with high (vs. low) cognitive effort investment with increasing payoff in the *n*-back and flanker task.

Exploratorily, we also report the 3-way interaction between cognitive effort investment, demand and payoff for all effort indices as well as both tasks and examine the presented hypotheses (H1—H4) regarding the two subdimensions of cognitive effort investment, i.e., cognitive motivation and effortful self-control. We hope to add value to the scientific community for future studies or meta-analyses with these additional exploratory analyses.

## Methods

We report how we determined our sample size, all data exclusions (if any), all manipulations, and all measures in the study [cf. 37]. We provide all data and analysis code to reproduce our results at https://osf.io/p58ub.

## Participants

The final sample included 148 participants (62% female, age: 24.1 ± 4.0 years) of whom 97% had a matriculation standard and 86% being students. All participants had normal or corrected-to normal-vision. All participants gave written informed consent in accordance with the Declaration of Helsinki on arrival at both measurement occasions. Participants received 10 € per hour or course credit for participation as well as a varying gain from the reaction time paradigms. The ethics board of the TU Dresden approved the study protocol, *reference number*: EK 3012016.

## Procedure

Considering feasibility and time constraints, we strived for the largest possible sample size of $N = 150$ participants. We performed sensitivity analyses with Optimal Design with Empirical Information (OD+) [38] in order to detect effect size $\delta$, with $N$ being the sample size, $n$ the number of trials per condition and $\rho$ the intra-class correlation. This target sample size would enable us to detect an effect size $\delta = 0.16$ for the $n$-back task ($\alpha = 0.05$, $n = 72$) and $0.15 < \delta < 0.18$ for the flanker task ($\alpha = 0.05$, $20 < n < 136$) with an assumed $\rho = 0.10$ at power $1\text{-}\beta = .80$. We recruited $N = 210$ participants via online and offline advertisements at Dresden's universities and the universities' participant recruiting platform. In a standardized telephone interview, we assessed criteria for inclusion (i.e., aged between 18 and 38, fluent in German, right-handed, normal or corrected-to-normal vision) and exclusion (i.e., psychological, psychiatric or neurological pre-existing conditions, regular illegal or excessive legal drug intake, regular intake of medication affecting cognitive performance, other conflicting reasons, dreadlocks, red-green-vision impairment, previous participation) via collective and single questions. Experimental sessions took place at two time points (henceforth referred to as t1 and t2) with one week in between (mean: 7.4 days, range: 6–18 days). We kept weekday (95% same) and daytime (85% on time, range: 2 h earlier– 2.5 h later) similar where possible. In general, both sessions followed the same procedure except for the color vision test and the cognitive task battery, which were only part of t1. After preparation of EEG and eyetracking, participants performed two computer tasks in randomized order. After the first task, they answered personality questionnaires. Fig 3 displays the whole procedure. The present study focusses on the $n$-back and the flanker task, the personality questionnaires of t1 as well as the cognitive

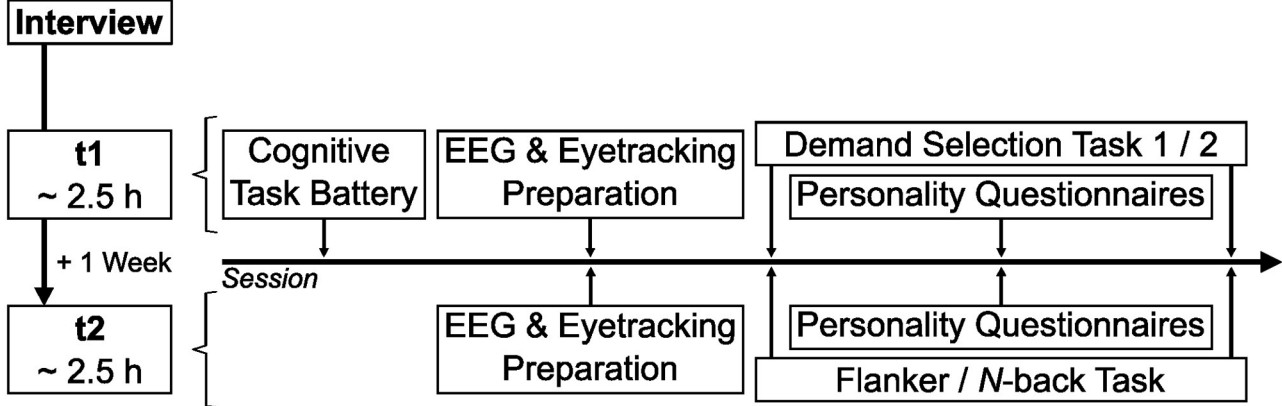

**Fig 3. Schematic illustration of the study procedure.**

task battery, which we discuss in more detail. Of $N = 170$ eligible participants, 22 were excluded due to conflicting time schedules, dyschromatopsia, technical problems or sleep deficits. The final sample allowed us to detect effect sizes between $0.07 < \delta < 0.35$ for the *n*-back task ($146 < N < 148$, $\alpha = 0.05$, $n = 72$, $0.01 < \rho < 0.54$) and $0.08 < \delta < 0.27$ for the flanker task ($144 < N < 147$, $\alpha = 0.05$, $20 < n < 136$, $0.02 < \rho < 0.30$) at power $1\text{-}\beta = .80$ (see S1 Fig).

## Tasks and measures

**Self-report measures.** Besides some sociodemographic details, we employed four questionnaires to access dispositional cognitive effort investment and asked participants to rate their perceived task load.

*Sociodemography*. We assessed age, sex, graduation, current occupation, use of visual aid, and visual acuity using multiple-choice questions.

*Need for cognition*. The 16-item short version of the German Need for Cognition Scale [39] assessed need for cognition. Responses to each item (e.g., "I really enjoy a task that involves coming up with new solutions to problems." [40]) were recorded on a 7-point Likert scale ranging from -3 (disagree strongly) to +3 (agree strongly). The scale shows comparably high internal consistency with Cronbach's $\alpha > .80$ [39, 41], and a retest reliability of $r_{tt} = .83$ across 8 to 18 weeks [42].

*Intellect*. To measure intellect, we employed the 24-item Intellect Scale by Mussel [9]. It assesses two intellectual processes (seek and conquer) and three intellectual operations (think, learn, and create). Items (e.g., "I enjoy solving complex problems" for the seek/think facet or "When I'm developing something new, I can't rest until it's completed" for the facet of create/conquer) are rated on a 7-point Likert scale ranging from -3 (disagree strongly) to +3 (agree strongly). Internal consistency is high (Cronbach's alpha = .94 for the total Intellect score and $\geq .86$ for the six facets [9]), and 1-year retest reliability is acceptable ($r_{tt} = .73$ for the total Intellect score and $\geq .58$ for the six facets; Mussel, P., personal communication, June 24, 2020).

*Self-control*. Self-control was measured using the short form of the German Self-Control Scale (SCS-K-D) [43] that comprises 13 items (e.g., "I am able to work effectively toward long-term goals") with a 5-point Likert scale ranging from -2 (disagree strongly) to +2 (agree strongly). The scale shows comparably high reliability (Cronbach's alpha ~ .80, 7-week retest reliability rtt = .82 [44]).

*Effortful control*. The respective scale of the German ATQ [10] assessed effortful control. It comprises 19 items on executive control in everyday life. Responses to items (e.g., "If I think of something that needs to be done, I usually get right to work on it.") are given on a 7-point Likert scale from -3 (disagree strongly) to +3 (agree strongly). Internal consistency of the scale is acceptable (Cronbach's alpha = .74 [10]), and 5-week retest reliability is high with $r_{tt} = .80$ [45].

*Cognitive effort investment*. We performed a confirmatory factor analysis (CFA) to derive factor scores for cognitive effort investment, cognitive motivation and effortful self-control according to the procedure described in Kührt et al. [8]. The second-order latent variable cognitive effort investment integrates the two first-order latent variables cognitive motivation and effortful self-control. We calculated cognitive motivation from overall test scores of indicator variables need for cognition and intellect; and effortful self-control from overall test scores of indicator variables self-control and effortful control.

*NASA-TLX*. We employed a measure of perceived task load, the NASA Task Load Index [19], on the cognitive task battery, the flanker task and three demand levels of the *n*-back task. In the NASA-TLX, participants evaluate their subjective perception of mental, physical and temporal demands of a particular task, as well as their performance, effort and frustration on a

20-point scale for each dimension. In its original, the NASA-TLX also requires comparisons of two dimensions each. For this study, we relinquished the comparison due to time restrictions. In order to obtain a uniform measure, we performed a principal component analysis with all scales. The theoretically assumed one-factor solution for perceived task load was confirmed by both parallel-analysis and scree-Test. All scales loaded highly on this factor; standardized loadings ranged from 0.43 for physical to 0.82 for effort, frustration and time. The factor scores of the unrotated one-factor solution served as a measure of *perceived task load*.

**Cognitive task battery.** The following tasks assessed markers for relevant cognitive abilities and intelligence that may affect task processing:

*Digit Symbol Substitution Test.* This test belongs to the German version of the WAIS-III [46] and assesses the speed of information processing. The digits 1 to 9 have to be substituted as fast as possible with symbols according to a substitution scheme printed on top of the page. It contains seven rows of 20 digits each being processed within two minutes including eight training trials. The outcome measure is the number of correctly substituted symbols and can take values of up to 133.

*Digit Span Backwards Test.* This test also belongs to the German version of the WAIS-III [46] and measures working memory capacity. It consists of seven tasks with two trials each that comprise an increasing number of digits (2–8). The participant listens to the digits and repeats them in reverse order. The task stops, if the second reproduction attempt is incorrect. The outcome measure is the number of the last correctly performed trial (1–14) minus the number of errors before the last correct trial.

*Trail Making Test A and B.* Two versions of the Trail Making Test [see e.g., 47] were used to examine mental speed (versions A and B) and task shifting ability (version B only). In version A, the participants are asked to connect 25 numbers scattered across a sheet of paper in ascending order. In version B, participants connect numbers and letters in alternating order (i.e., 1-A-2-B . . .). The outcome measure is the time in seconds for completion of each version.

*Cognitive abilities.* Outcome measures of Trail Making Test A and B were log-transformed and inverted in order to display a positive correlation with performance. These measures as well as the outcome measures of the Digit Symbol Substitution Test and the Digit Span Backwards Test were z-transformed and entered into a principal component analysis. The factor scores of the unrotated one-factor solution served as marker for *cognitive abilities*.

*Multiple-Choice Vocabulary Test.* We assessed crystallized intelligence by means of the Multiple-Choice Vocabulary Test [48]. This test consists of 37 lists composed of one real German word and four pseudo words. The participants' task was to identify the real word. The outcome measure is the number of correct answers and served as marker for crystalized intelligence.

**Behavioral tasks.** Participants performed two typical cognitive control tasks with varying demand and payoff levels. The order of the tasks was counterbalanced across participants.

*N-back.* Participants completed an *n*-back task (Fig 4) with varying levels of difficulty (0-back, 1-back, 2-back) as specified by Mussel et al. [13]. For the 0-back task, participants were instructed to press the left arrow key whenever the presented letter was an "X" and the right arrow key otherwise. For the 1-back task participants were instructed to press the left arrow key whenever the current letter was identical to the letter presented one serial positions before; two serial positions before for the 2-back task. In each trial of the *n*-back task, a single letter was presented in the center of the screen until the participant pressed the response button or until the response time limit (2000 ms) was reached. Between any two letters, a fixation cross was presented for 500 ms. During the fixation period participants received performance feedback for the subsequent trial.

Before starting the task proper, participants completed (and repeated, if necessary) 20 training trials for each difficulty level until they completed at least 85% of trials correctly. The task

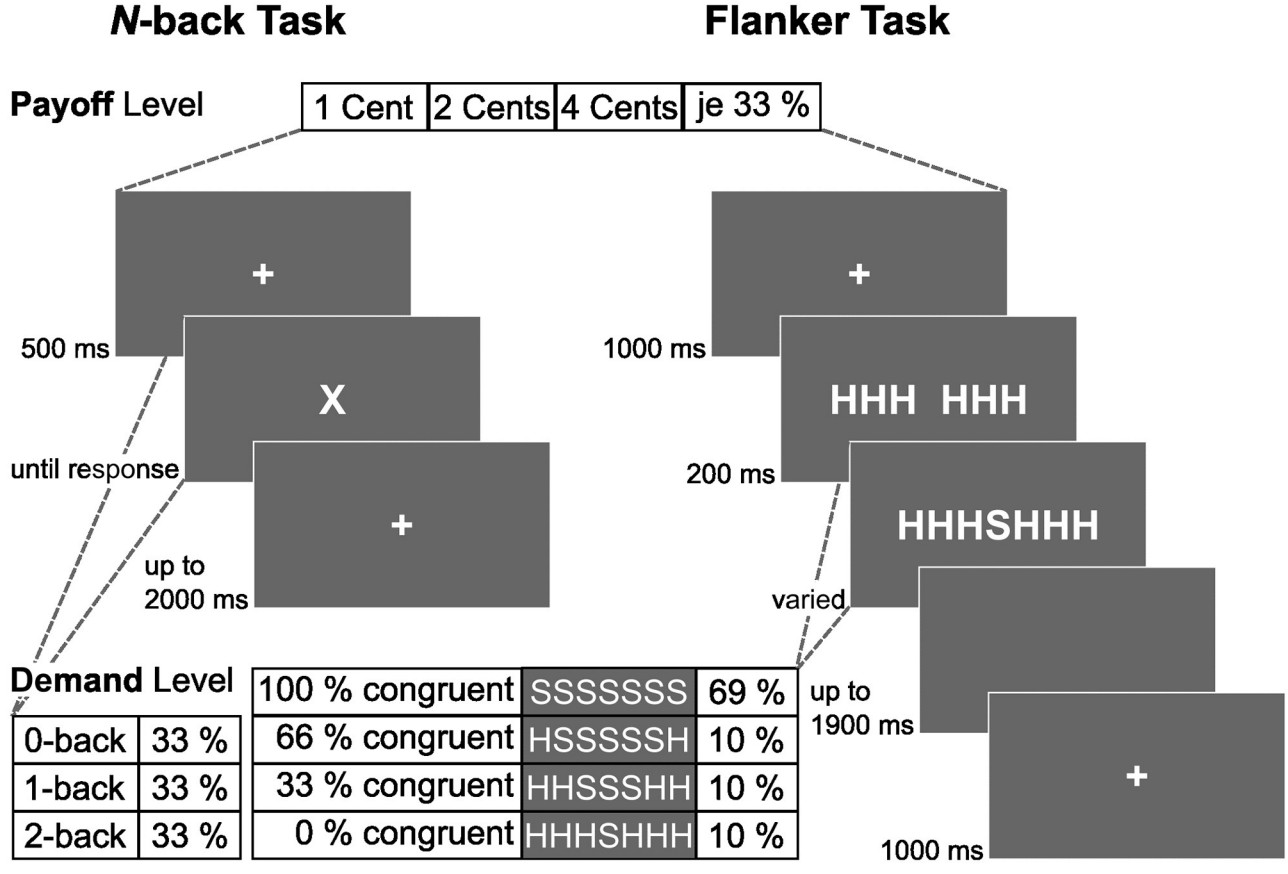

**Fig 4. Schematic trial procedure of the *n*-back and flanker task.**

comprised nine blocks of 72 trials with varying demand and payoff between blocks (i.e., each combination once per condition). The blocks commenced with a display that indicated the current *n*-back level (0, 1, 2) and the payoff that participants would receive for a correct trial (0.01, 0.02, 0.04 EUR). Participants also received information on their overall performance at the end of each block.

*Flanker task*. Participants completed a flanker task (Fig 4) with varying levels of distraction based on the specifications provided by Forster et al. [21]. Each stimulus was composed of letters "H" and "S". Participants were instructed to press the left arrow key whenever the central letter was an "H" and the right arrow key whenever the central letter was an "S". The central letter was flanked by either compatible (same as the central letter) or incompatible letters (different than the central letter). The proportion of congruent flanking letters varied to yield four demand levels (1,2,3,4): 100% compatible (e.g., SSSSSSS; 69% of the trials), 67% compatible (e.g., HSSSSSH; 10% of the trials), 33% compatible (e.g., HHSSSHH; 10% of the trials), and 0% compatible (e.g., HHHSHHH; 10% of the trials). In each trial, flanker letters were presented 200 ms before the target stimulus appeared on screen and both remained visible either until the participant pressed the response button or until the response time limit (1200 ms) was reached. The trial was then followed with a period of 700 ms during which the screen was blank and a subsequent fixation period of 1000 ms during which participants received feedback.

Before starting the task proper, participants completed (and repeated, if necessary) 20 training trials until they completed at least 85% of trials correctly. The task consisted of six blocks of 98 trials with demand varying randomly within blocks and payoff between blocks (i.e., two blocks per payoff level). We randomized the order of payoff blocks to avoid sequence effects. Each block commenced with a display that indicated the payoff that participants would receive for a correct trial (0.01, 0.02, 0.04 EUR) and ended with information on the overall performance.

Further measures not relevant to this study were the Ishihara's Test for Colour Deficiency [49] (t1, plates 3 or 17 and 33), two variants of the Demand Selection Task (DST) as established by Kool et al. [3] (t1), short form of personality questionnaires to access cognitive effort investment (t2), as well as follow-up questions on previous experience with the applied paradigms. A detailed summary of all measures in the study are openly accessible at https://osf.io/p58ub.

**RT preprocessing.** We considered reaction times < 100 ms as false alarm, as they are faster than simple reaction time, i.e. time needed to respond to the stimuli's presence [50]. Thus, such responses were excluded from analysis. We addressed common reaction time preprocessing issues by analyzing the data with linear mixed models [51].

**EEG recording and preprocessing.** We recorded EEG during *n*-back and flanker task from 31 gel-filled Ag-AgCl electrodes applied to an M1 Easycap (EASYCAP GmbH, Herrsching, Germany) according to the 10–20 system. Additionally, we tracked electrooculogram (EOG), left mastoid as reference and ground. We sampled the signal at 500 Hz and amplified it using BrainAmp DC (Brain Products GmbH, Gilching, Germany). DC offset correction was performed automatically at a threshold of 70%. We kept all impedances below 10 kΩ. BrainVision Recorder 1.23 (Brain Products GmbH, Gilching, Germany) logged the EEG signal. We preprocessed the recording using BrainVision Analyzer 2.2 (Brain Products GmbH, Gilching, Germany). We linearly derived horizontal and vertical EOG from F9/F10 and EOG/Fp1, respectively. We applied a second order Butterworth filter with 0.15 Hz low cutoff, 40 Hz high cutoff and 50 Hz notch filter. Occular correction followed the Gratton and Coles method [52] as it performs comparatively well in BrainVision Analyzer. Next, we segmented the data to -1300 to 3100 ms relative to target stimuli (*n*-back)/ flanker stimuli (flanker). This interval length buffers edge artifacts that usually occur during wavelet transformation [53]. Artifacts in the channels of interest (Cz, FCz, Fz, CPz, Pz) were automatically inspected in the time window of -400–1900 ms (*n*-back)/ 2100 ms (flanker) and rejected if exceeding the maximal allowed voltage step (50 µV/ms) or the maximal allowed amplitude (± 100 µV). We re-referenced all channels to the grand average. Within these steps, we marked bad segments without removing them.

*Event-related potentials*. We cut the segments to -150 to 1900 ms (*n*-back) / 2100 ms (flanker). Segments were then baseline corrected (-150 –-50 ms relative to target stimuli (*n*-back)/ flanker stimuli (flanker)). We exported the generic data at channels Fz, FCz, Cz, CPz, and Pz. N2 amplitude was defined as mean voltage at Fz, FCz, and Cz in the time window 310 ± 20 ms from target onset. P3 amplitude was defined as mean voltage at Cz, CPz, and Pz in the time window 400 ± 20 ms from target onset. Inspection of the grand mean revealed a good mapping of the respective peaks through the pre-defined time windows. We calculated amplitudes for each trial.

*Time-frequency analysis*. We applied a continuous wavelet transformation with complex Morlet function. The frequency ranged from 1 to 20 Hz and 50 frequency layers were calculated with logarithmic steps. For wavelet normalization, we selected instantaneous amplitude. Morlet parameter was set to 5. Layers were baseline corrected (-400 –-100 ms). The baseline interval covers a bit more than one cycle of the lowest frequency in the theta band as Cohen [53] recommended. We exported wavelet data for the layers 24 to 33 (corresponding to 4.08–

7.07 Hz) for early (0–650 ms) and late processes (650–1900 ms) relative to target onset as sum scores for channels Cz, FCz and Fz. Early and late FMθ power were calculated as means of these channels for each trial.

Trials with missing data, marked as bad segment or with RTs <100 ms were excluded from statistical analysis.

**Pupil recording and preprocessing.** Pupil size and gaze data were recorded monocularly (91% right eye) with an EyeLink 1000 Plus system (SR Research, Mississauga, Ontario, Canada) in remote desktop mount configuration with a sampling rate of 500 Hz. The distance between the eye camera and participants' eye was about 60 cm. A chin rest was used to keep the distance constant across the experimental session (note: participants were required to only slightly touch the chin rest but not to lean onto it, in order to avoid muscular artefact in the EEG recordings). A 9-point calibration and validation routine was performed at the beginning of each experimental task, after (planned) breaks within experimental sessions and in cases when measurement errors (lost pupil) occurred. A drift correction was performed before each experimental block or when a re-calibration was needed within a block. Pupil size is reported in arbitrary units. The value refers to the number of pixels recognized as pupil in each image of the eye tracker video stream.

Preprocessing of pupil size data based on the guidelines of Kret and Sjak-Shie [54] and was done with R Software for Statistical Computing (versions 4.1.0 and 4.1.2) for Windows using RStudio (versions 1.4.1717 and 2022.02.0). We used the packages *eyelinker* (version 0.2.1) [55] and *gazer* (version 0.1) [56]. First, we filtered raw data for invalid pupil size samples. We set non-positive pupil sizes and off-screen data to *NA*, as well as pupil sizes detected within blinks (blink artifacts) including the surrounding saccades (edge artifacts). We calculated pupil size change as normalized dilation speed and set pupil sizes to *NA* if the related speed exceeded the threshold of mean speed*16 (dilation speed outliers). Next, we generated a trend-line by interpolating (cubic spline) and smoothing the remaining pupil sizes. We set pupil sizes to *NA* if the deviation from this trend-line exceeded the threshold of median deviation + 16*median absolute deviation (absolute trend-line deviation outliers). We repeated this procedure one more time on the remaining pupil data. A sparsity filter set clusters smaller than 50 ms to *NA* that followed a gap lager than 40 ms (temporarily isolated samples). Second, preprocessing the valid samples included up-sampling to 1000 Hz and interpolating the new data points by type cubic-spline, but all previously excluded or missing values were still treated as *NA* as Mathot et al. [57] recommend. Third, we sectioned the data to 0 to 1900 ms (*n*-back) / 2100 ms (flanker) relative to target onset. Fourth, for each trial, we calculated baseline as the median of the first 10 ms and subtracted it from the interval as recommended [57]. If the baseline resulted with no valid pupil data, we set this trial to *NA*. Data was then aggregated as mean pupil size across intervals for each trial. Trials with missing data, set to *NA* or with RTs < 100 ms were excluded from statistical analyses. This affected an average of 2% of trials per participant (*M* = 13 trials, ranging from 0 to 147) in the *n*-back task and an average of 3% of trials per participant (*M* = 15 trials, ranging from 0 to 538) in the flanker task.

## Statistical analyses

We performed all statistical analyses with R Software for Statistical Computing (versions 4.1.0 and 4.1.2) for Windows using RStudio (versions 1.4.1717 and 2022.02.0). In particular, we used the packages *lmerTest* (version 3.1–3) [58], *interactions* (version 1.1.5) [59], and *MuMIn* (version 1.43.17) [60]. As some of the trait variables deviated from univariate normality (Shapiro–Wilk tests, p $\geq$ 0.20), all trait variables were normalized using Blom's formula ((r − 3/8)/ (n + 1/4), with r being the rank of observations and n the sample size) [61] and standardized.

The normalized variables did not deviate from univariate normality (Shapiro–Wilk tests, $p \geq 0.20$). Given the multilevel structure of the data due to repeated measures, the unbalanced design as well as missing values, linear mixed models best fit the structure of the data. As response coherence across the different effort indices cannot generally be assumed [e.g., 62], we refrained from corrections for multiple testing.

**Linear mixed models.**    In order to test hypotheses H1, H3 and H4, we estimated several linear mixed models (LMM) on the single trial data. Each model followed the formula:

$$effort\ index \sim demand * payoff * cognitive\ effort\ investment + (demand * payoff \mid subject)$$

with *effort index* standing for the respective effort index, i.e. subjective load, reaction time, FMθ power early, FMθ power late, N2 amplitude, P3 amplitude, or pupil dilation; *demand* (level 1, continuous), *payoff* (level 1, continuous) and *cognitive effort investment* (level 2, continuous) being predictor variables. We controlled for *correct* (level 1, factors correct/ incorrect) and *post-correct responses* (i.e., the preceding trial was correct; level 1, factors post-correct/ post-incorrect), *block* (level 1, continuous), *cognitive abilities* (level 2, continuous) and *crystalized intelligence* (level 2, continuous) by including them as additional predictors into our models. Additionally, responses after an individual RT deadline (i.e., median of correct practice trial RTs) in the flanker task were considered late (*late responses*: level 1, factors late/ not late), as speed pressure may yield stronger conflict effects [63] and was considered before in the study of Forster et al. [21]. We centered all predictors in accordance with the recommendations of Enders and Tofighi [64] in order to obtain interpretable parameter estimates. As the focus of this study lies on interaction effects, we centered all level 1 predictors within person, i.e., group mean centering. We centered all level 2 predictors at the grand mean. We applied orthogonal sum-to-zero contrasts in order to get meaningful Type-III tests. We fitted the models with restricted maximum likelihood (REML).

*Null model.* We estimated the null model in order to calculate the intra-class correlation. It serves as an effect size measure for random effects and displays the proportion of variance that is explained by differences between persons.

*Random slopes model.* We estimated the maximal models and, if necessary, adjusted the optimization algorithm to improve model fit. We visually inspected the residuals of all models.

**Generalized linear mixed model.**    As accuracy follows a binomial distribution, we estimated a generalized linear mixed model (GLMM) for accuracy. In general, the structure followed the one of the linear mixed models. The model was fitted with maximum likelihood (Laplace Approximation) and logistic link function. We visually inspected model fit.

**Exploratory analyses.**    We additionally estimated all LMM and GLMM models with cognitive motivation and effortful self-control in place of cognitive effort investment. All specifications outlined above remained the same for these additional analyses. We provide the full results of these models in S1 Table and report them here only if significant or if results of first- and second-order factor differ.

**Effect sizes.**    We provide conditional pseudo $R^2$ for each model as additional effect size measure. It indicates the variance explained by the whole model.

**Simple slopes analysis.**    To evaluate the interaction effects, we performed simple slopes analysis. False discovery rate was adjusted for each analysis. We provide the simple slopes analysis results in S2 Table and report them only for significant cognitive effort investment × payoff or demand interactions here. We calculated Johnson-Neyman intervals (see S2 Fig for Johnson-Neyman plots). Interaction plots are based on model parameters. S3 Fig displays plots of all interactions, while in the results section we focused on plots for significant interactions between cognitive effort investment × demand or payoff.

## Results

### Manipulation check: Effort indices reflect levels of demand and partly of payoff

The LMMs and GLMMs revealed significant main effects of demand for perceived task load, reaction time, accuracy, early FMθ power, N2 and P3 amplitude in both tasks, all $p \leq .011$, and late FMθ power, pupil dilation in the flanker task, $p < .036$ (cf. Tables 2–9).

The models revealed significant main effects of payoff for reaction time, $p = .036$, P3 amplitude and pupil dilation, both $p \leq .001$, in the flanker task (cf. Tables 3–9).

### Cognitive abilities and crystallized intelligence: Almost no impact

Table 1 displays the main effects of cognitive abilities and crystallized intelligence that we have controlled for in all LMMs and GLMMs. While cognitive abilities affected two effort indices in the flanker task and four in the *n*-back task, crystallized intelligence proved significant for only one effort index in each task.

### Perceived task load: No relation to cognitive effort investment

Table 2 provides the correlation analysis and Table 3 displays the full results of the LMM.

**N-back.** The intra-class correlation indicated that differences between subjects explain 54% of the variance. The whole LMM explained 90% of the variance (conditional pseudo $R^2$). It revealed neither a main effect for cognitive effort investment nor an interaction effect for

**Table 1. Main effects of cognitive abilities and crystallized intelligence in *n*-back and flanker task.**

| Parameter | N-back | | Flanker | |
|---|---|---|---|---|
| | **Beta** | **p-value** | **Beta** | **p-value** |
| *Cognitive Abilities* | | | | |
| Perceived Task Load | -0.07 | .268 | | |
| Reaction Time | -7.19 | .110 | -6.10 | .104 |
| Accuracy | 0.26 | < .001*** | 0.04 | .577 |
| Early FMθ Power | 0.31 | .014* | 0.23 | .007** |
| Late FMθ Power | -0.01 | .838 | -0.06 | .390 |
| N2 Amplitude | 0.45 | .008** | 0.54 | .005** |
| P3 Amplitude | 0.53 | .001** | 0.22 | .201 |
| Pupil Dilation | -1.85 | .455 | -4.77 | .091 |
| *Crystallized Intelligence* | | | | |
| Perceived Task Load | 0.00 | .893 | | |
| Reaction Time | 0.75 | .601 | 1.31 | .273 |
| Accuracy | -0.01 | .739 | 0.04 | .048* |
| Early FMθ Power | -0.08 | .045* | -0.02 | .371 |
| Late FMθ Power | -0.02 | .148 | 0.01 | .662 |
| N2 Amplitude | 0.05 | .358 | -0.09 | .159 |
| P3 Amplitude | -0.05 | .338 | -0.04 | .506 |
| Pupil Dilation | 1.06 | .187 | 0.36 | .692 |

*Note.* $N \geq 144$.

***$p < .001$,

**$p < .01$,

*$p < .05$

**Table 2. Correlation matrix for perceived task load in the cognitive task battery and flanker task.**

|  |  | 1 | 2 | 3 | 4 |
|---|---|---|---|---|---|
| 1 | CEI | - |  |  |  |
| 2 | Cognitive Motivation | .88*** | - |  |  |
| 3 | Effortful Self-Control | .84*** | .47*** | - |  |
| 4 | **Cognitive Task Battery** | -.12 | -.01 | -.21 | - |
| 5 | **Flanker** | -.14 | -.11 | -.13 | .23* |

*Note.* N = 147.

\*\*\**p* < .001,

\*\**p* < .01,

\**p* < .05.

CEI = cognitive effort investment.

cognitive effort investment × demand, all *p* > .05. Comparable results were obtained when performing this analysis with cognitive motivation, however when performing with effortful self-control it produced a main effect, *β* (SE) = -0.21 (0.09), *p* = .014 (S1 Table).

**Flanker and cognitive task battery.** Neither cognitive effort investment nor its components were significantly correlated with perceived task load reported after the cognitive task battery nor in the flanker task.

## Reaction time: Individuals with high cognitive effort investment respond equally fast or slower as payoffs increase

Table 4 displays the full results of the LMMs and simple slopes analyses for the significant cognitive effort investment × payoff interactions.

**N-back.** The intra-class correlation indicated that differences between subjects explain 16% of the variance. The whole LMM explained 38% of the variance (conditional pseudo $R^2$). It revealed no main effect for cognitive effort investment, and no interaction effect for cognitive effort investment × demand, both *p* > .05. The interaction cognitive effort investment × payoff was significant, *p* = .003 (Fig 5A). The simple slopes analysis yielded significant slopes of payoff only for high levels of cognitive effort investment, *p* < .01. Hence, reaction times only increased in individuals with high cognitive effort investment due to increasing payoff, whereas reaction times remained unchanged in individuals with low or

**Table 3. Results for perceived task load analysis in *n*-back task.**

| Parameter | N-back | | | |
|---|---|---|---|---|
|  | **Beta** | **SE** | **_p_-value** | **Random Effects (SD)** |
| Intercept | -0.42 | 0.06 | < .001*** | 0.73 |
| Demand | 0.45 | 0.03 | < .001*** | 0.30 |
| CEI | -0.30 | 0.16 | .067 |  |
| CEI × Demand | -0.05 | 0.08 | .561 |  |

*Note.* $N_{n\text{-back}}$ = 148, $N_{flanker}$ = 147, $N_{cognitive\ task\ battery}$ = 147.

\*\*\**p* < .001,

\*\**p* < .01,

\**p* < .05.

CEI = cognitive effort investment.

**Table 4. Results for reaction time analysis in *n*-back and flanker task.**

| Parameter | | N-back | | | | Flanker | | | |
|---|---|---|---|---|---|---|---|---|---|
| | | Beta | SE | *p*-value | Random Effects (SD) | Beta | SE | *p*-value | Random Effects (SD) |
| Intercept | | 547.66 | 6.30 | < .001*** | 76.35 | 348.76 | 4.14 | < .001*** | 50.11 |
| Demand | | 83.40 | 4.94 | < .001*** | 59.64 | 13.43 | 0.58 | < .001*** | 6.55 |
| Payoff | | 0.37 | 0.77 | .635 | 08.09 | -0.78 | 0.37 | .036* | 4.09 |
| CEI | | 0.17 | 16.62 | .992 | | 2.85 | 10.99 | .796 | |
| Demand × Payoff | | 1.05 | 1.18 | .374 | 13.03 | -0.15 | 0.16 | .340 | 0.76 |
| CEI × Demand | | 2.28 | 12.97 | .861 | | 1.58 | 1.50 | .297 | |
| CEI × Payoff | | 6.09 | 2.03 | .003** | | 2.77 | 0.97 | .005** | |
| CEI × Demand × Payoff | | 5.38 | 3.09 | .083 | | -0.65 | 0.42 | .129 | |
| | Value of CEI | Beta (SE) | 95% CI | Conditional Intercept Beta (SE) | | Beta (SE) | 95% CI | Conditional Intercept Beta (SE) | |
| Payoff | - 1 SD | -1.95 (1.09) | [-4.10, 0.19] | 547.60 (8.93) | | -1.84 (0.52)*** | [-2.86, -0.81] | 347.67 (5.88) | |
| | Mean | 0.37 (0.77) | [-1.15, 1.89] | 547.66 (6.30) | | -0.78 (0.37)** | [-1.51, -0.06] | 348.76 (4.14) | |
| | + 1 SD | 2.69 (1.09)** | [0.54, 4.83] | 547.72 (8.93) | | 0.27 (0.52) | [-0.76, 1.30] | 349.84 (5.88) | |

*Note.* $N_{n\text{-back}}$ = 148, $N_{flanker}$ = 147.

***$p$ < .001,

**$p$ < .01,

*$p$ < .05.

CEI = cognitive effort investment.

medium cognitive effort investment. The exploratory models produced comparable results (S1 Table), with cognitive motivation × payoff, $\beta$ (SE) = 3.12 (1.03), $p$ = .003 and effortful self-control × payoff, $\beta$ (SE) = 2.20 (1.09), $p$ = .045.

**Flanker.** The intra-class correlation indicated that differences between subjects explain 30% of the variance. The whole LMM explained 64% of the variance (conditional pseudo $R^2$).

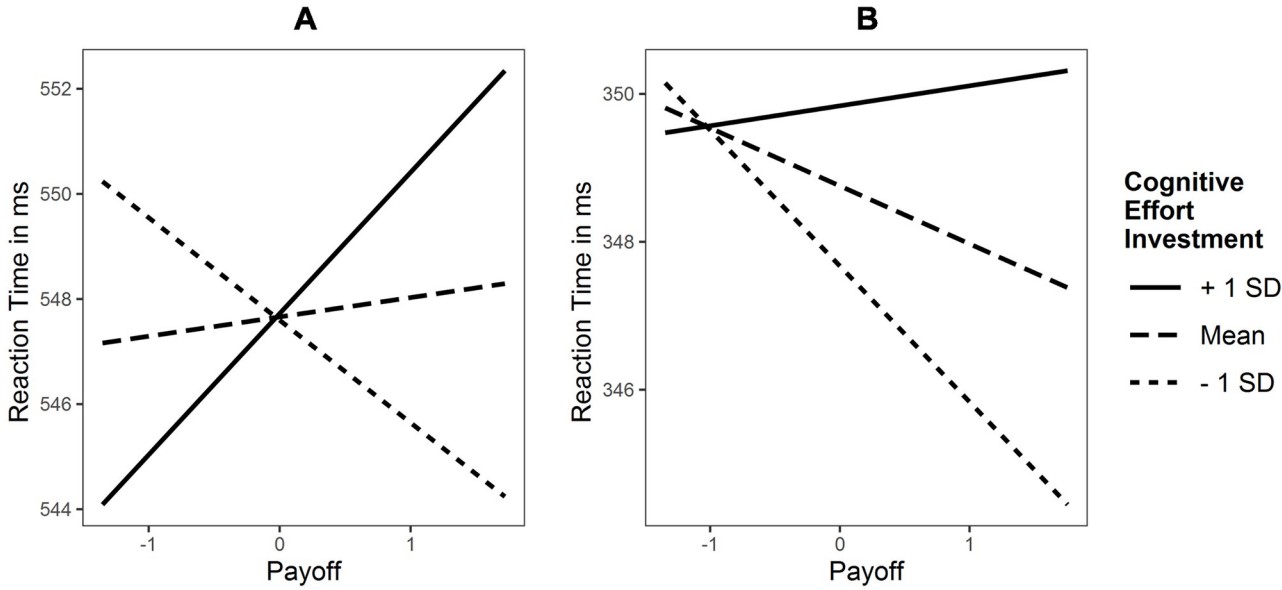

**Fig 5. Interaction plots of cognitive effort investment × payoff interaction for reaction time in the (A) *n*-back and (B) flanker task.**

It revealed no main effect for cognitive effort investment, and no interaction effect for cognitive effort investment × demand, all $p > .05$. The interaction cognitive effort investment × payoff was significant, $p = .005$ (Fig 5B). The simple slopes analysis yielded significant slopes of payoff for low, $p < .001$, and mean levels of cognitive effort investment, $p < .01$, but not for high, $p > .05$. Thus, reaction times decreased in individuals with low and medium cognitive effort investment in response to a payoff increase, but were unaffected in individuals with high cognitive effort investment. The results for the exploratory models were comparable (S1 Table), with cognitive motivation × payoff, $\beta$ (SE) = 1.25 (0.50), $p = .013$ and effortful self-control × payoff, $\beta$ (SE) = 1.20 (0.52), $p = .022$.

## Accuracy: Main effect of cognitive effort investment in the $n$-back task

Table 5 displays the full results of the GLMMs.

**N-back.** The whole GLMM explained 26% of the variance (conditional pseudo $R^2$). It revealed a main effect for cognitive effort investment, $p = .019$. There were no interaction effects for cognitive effort investment × demand and cognitive effort investment × payoff, both $p > .05$. Exploratory analyses revealed only a main effect of effortful self-control, $\beta$ (SE) = 0.20 (0.08), $p = .018$, but not cognitive motivation, $\beta$ (SE) = 0.14 (0.08), $p = .094$, nor any other interaction effect (S1 Table).

**Flanker.** The whole GLMM explained 48% of the variance (conditional pseudo $R^2$). It revealed no main effect for cognitive effort investment, no interaction effect for cognitive effort investment × demand and cognitive effort investment × payoff, all $p > .05$. There were also no effects that involved first-order factors in the exploratory models (S1 Table).

## Early FMθ power: Steeper slope for individuals with high cognitive effort investment as demands increase

Table 6 displays the full results of the LMMs and simple slopes analysis for the significant cognitive effort investment × demand interaction in the flanker task.

**N-back.** The intra-class correlation indicated that differences between subjects explain 7% of the variance. The whole LMM explained 8% of the variance (conditional pseudo $R^2$). It

**Table 5. Results for accuracy analysis in $n$-back and flanker task.**

| Parameter | N-back | | | | Flanker | | | |
|---|---|---|---|---|---|---|---|---|
| | Beta | SE | $p$-value | Random Effects (SD) | Beta | SE | $p$-value | Random Effects (SD) |
| Intercept | 3.63 | 0.06 | < .001*** | 0.68 | 3.55 | 0.09 | < .001*** | 0.98 |
| Demand | -0.82 | 0.05 | < .001*** | 0.42 | -1.32 | 0.04 | < .001*** | 0.43 |
| Payoff | -0.01 | 0.02 | .779 | 0.01 | 0.03 | 0.03 | .183 | 0.14 |
| CEI | 0.37 | 0.16 | .019* | | 0.18 | 0.23 | .419 | |
| Demand × Payoff | 0.02 | 0.02 | .499 | 0.13 | 0.00 | 0.01 | .905 | 0.08 |
| CEI × Demand | 0.04 | 0.11 | .758 | | -0.03 | 0.10 | .773 | |
| CEI × Payoff | -0.02 | 0.04 | .607 | | 0.02 | 0.06 | .715 | |
| CEI × Demand × Payoff | -0.05 | 0.06 | .336 | | -0.02 | 0.03 | .487 | |

*Note.* $N_{n\text{-back}}$ = 148, $N_{flanker}$ = 147.

***$p < .001$,

**$p < .01$,

*$p < .05$.

CEI = cognitive effort investment.

**Table 6. Results for early frontal midline theta power analysis in *n*-back and flanker task.**

| Parameter | | N-back | | | | Flanker | | | |
|---|---|---|---|---|---|---|---|---|---|
| | | Beta | SE | *p*-value | Random Effects (SD) | Beta | SE | *p*-value | Random Effects (SD) |
| Intercept | | 2.27 | 0.13 | < .001*** | 1.57 | 2.41 | 0.14 | < .001*** | 1.67 |
| Demand | | -0.16 | 0.05 | .001** | 0.50 | 0.47 | 0.06 | < .001*** | 0.73 |
| Payoff | | 0.03 | 0.02 | .092 | 0.09 | -0.01 | 0.02 | .659 | 0.13 |
| CEI | | 0.57 | 0.35 | .104 | | 0.64 | 0.37 | .085 | |
| Demand × Payoff | | 0.00 | 0.02 | .843 | 0.14 | 0.00 | 0.02 | .951 | 0.13 |
| CEI × Demand | | 0.01 | 0.12 | .920 | | 0.38 | 0.17 | .026* | |
| CEI × Payoff | | -0.04 | 0.04 | .356 | | -0.02 | 0.05 | .667 | |
| CEI × Demand × Payoff | | -0.05 | 0.06 | .431 | | -0.11 | 0.05 | .027* | |
| | Value of CEI | | | | | Beta (SE) | 95% CI | | Conditional Intercept Beta (SE) |
| Demand | - 1 SD | | | | | 0.33 (0.09)*** | [0.15, 0.50] | | 2.16 (0.20) |
| | Mean | | | | | 0.47 (0.06)*** | [0.34, 0.59] | | 2.41 (0.14) |
| | + 1 SD | | | | | 0.61 (0.09)*** | [0.43, 0.79] | | 2.65 (0.20) |

*Note.* $N_{n\text{-back}}$ = 147, $N_{flanker}$ = 146.

***$p$ < .001,

**$p$ < .01,

*$p$ < .05.

CEI = cognitive effort investment.

revealed no main effect for cognitive effort investment, no interaction effect for cognitive effort investment × demand, and cognitive effort investment × payoff, all $p$ > .05. In line, exploratory analyses yielded no effects on cognitive motivation or effortful self-control (S1 Table).

**Flanker.** The intra-class correlation indicated that differences between subjects explain 7% of the variance. The whole LMM explained 12% of the variance (conditional pseudo $R^2$). It revealed no main effect for cognitive effort investment, and no interaction effect for cognitive effort investment × payoff, both $p$ > .05. The interaction cognitive effort investment × demand was significant, $p$ = .026 (Fig 6). The simple slopes analysis yielded significant slopes of demand for all levels of cognitive effort investment, all $p$ < .001. Thus, early FMθ power increased in all individuals with increasing demand, with steeper increase for higher levels of cognitive effort investment. The exploratory models produced comparable results (S1 Table), however the demand interaction was only significant for effortful self-control, $\beta$ (SE) = 0.18 (0.09), $p$ = .039, but not cognitive motivation × demand, $\beta$ (SE) = 0.15 (0.08), $p$ = .077.

## Late FMθ power: No effect of cognitive effort investment

Table 7 displays the full results of the LMMs.

**N-back.** The intra-class correlation indicated that differences between subjects explain 1% of the variance. The whole LMM explained 58% of the variance (conditional pseudo $R^2$). It revealed no main effect for cognitive effort investment, no interaction effect for cognitive effort investment × demand, and cognitive effort investment × payoff, all $p$ > .05. The same was true for the exploratory analyses yielding no effects on cognitive motivation or effortful self-control (S1 Table).

**Flanker.** The intra-class correlation indicated that differences between subjects explain 2% of the variance. The whole LMM explained 4% of the variance (conditional pseudo $R^2$). It revealed no main effect for cognitive effort investment, no interaction effect for cognitive effort

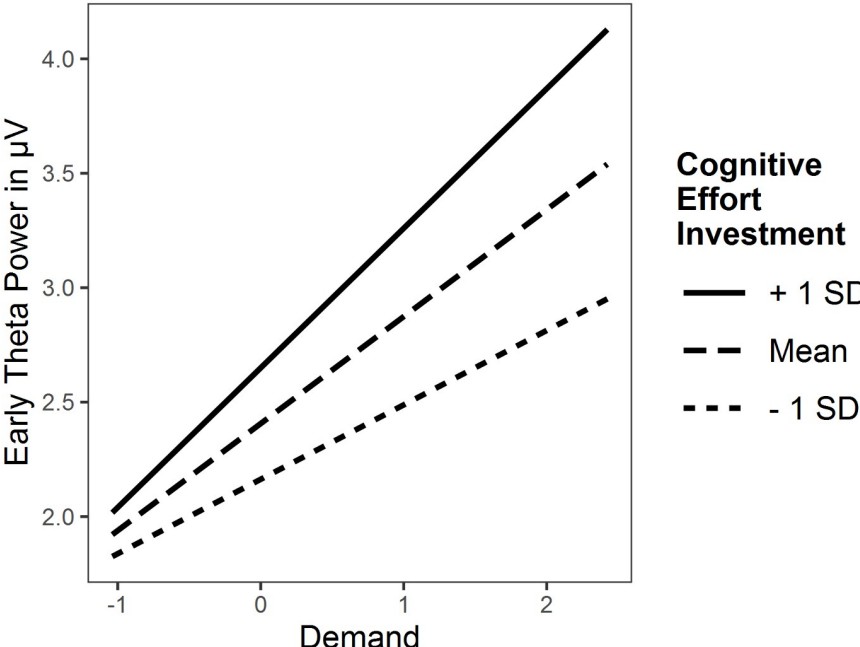

**Fig 6. Interaction plot of cognitive effort investment × demand interaction for early frontal midline theta power in the flanker task.**

investment × demand, and cognitive effort investment × payoff, all $p > .05$. Exploratory analyses yielded also no effects on the first-order factors (S1 Table).

## N2 amplitude: No effect of cognitive effort investment

Fig 7 shows the grand means of raw ERPs across participants and blocks at N2 relevant electrodes for both conditions and both tasks. Table 8 displays the full results of the LMMs.

**Table 7. Results for late frontal midline theta power analysis in *n*-back and flanker task.**

| Parameter | N-back | | | | Flanker | | | |
|---|---|---|---|---|---|---|---|---|
| | **Beta** | **SE** | ***p*-value** | **Random Effects (SD)** | **Beta** | **SE** | ***p*-value** | **Random Effects (SD)** |
| Intercept | 0.54 | 0.05 | $< .001^{***}$ | 0.59 | 0.55 | 0.08 | $< .001^{***}$ | 0.96 |
| Demand | 0.02 | 0.05 | .644 | 0.50 | -0.07 | 0.03 | $.036^{*}$ | 0.30 |
| Payoff | 0.01 | 0.02 | .550 | 0.08 | -0.02 | 0.02 | .187 | 0.10 |
| CEI | -0.20 | 0.14 | .156 | | -0.40 | 0.22 | .065 | |
| Demand × Payoff | 0.00 | 0.49 | .993 | 5.88 | -0.02 | 0.02 | .233 | 0.13 |
| CEI × Demand | 0.10 | 0.12 | .397 | | 0.01 | 0.08 | .943 | |
| CEI × Payoff | 0.02 | 0.04 | .656 | | 0.02 | 0.04 | .623 | |
| CEI × Demand × Payoff | -0.05 | 1.28 | .966 | | -0.03 | 0.05 | .479 | |

*Note.* $N_{n\text{-back}} = 147$, $N_{\text{flanker}} = 146$.

$^{***}p < .001$,

$^{**}p < .01$,

$^{*}p < .05$.

CEI = cognitive effort investment.

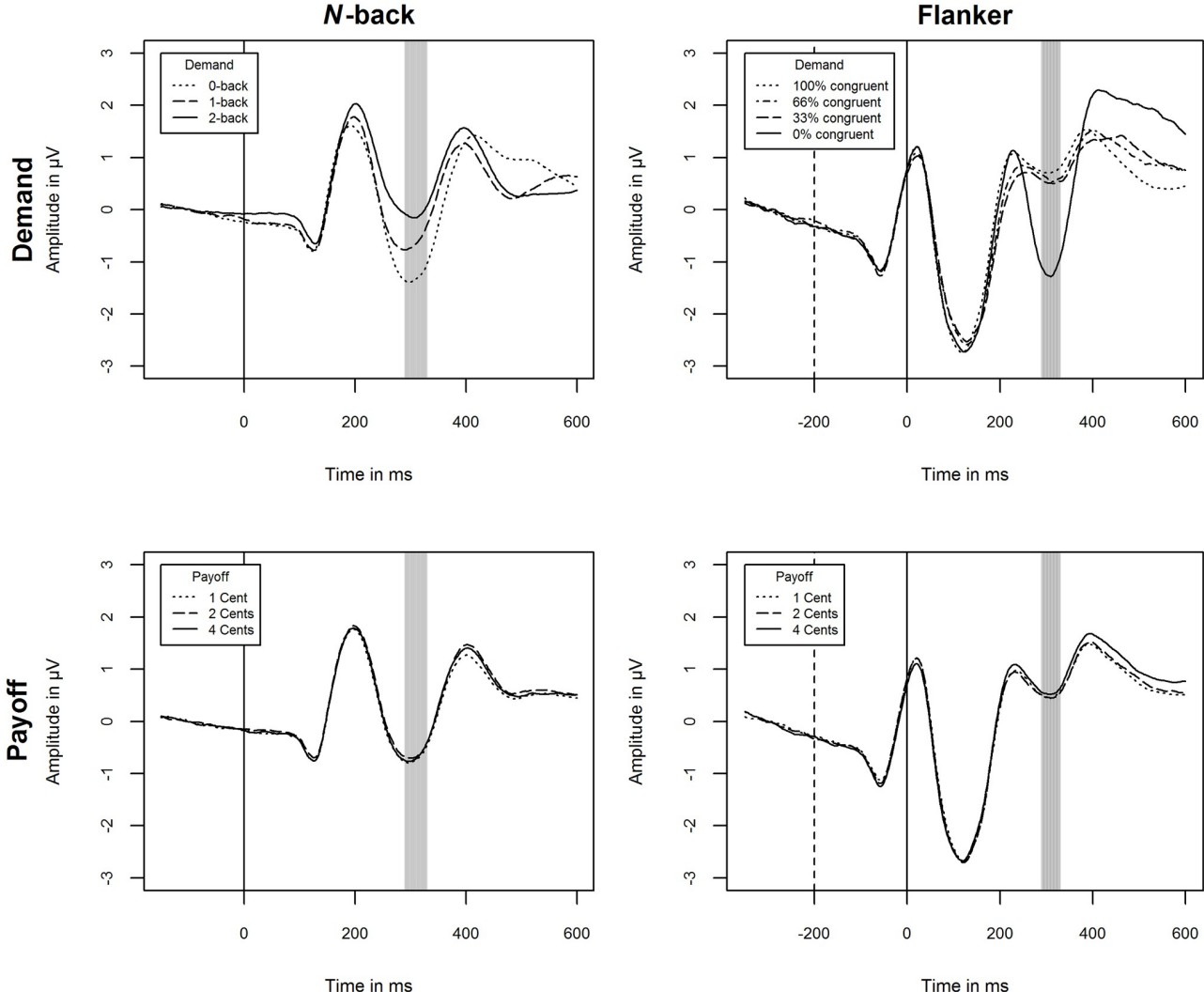

**Fig 7. Mean ERPs at Fz, FCz, and Cz by conditions demand and payoff for the *n*-back and flanker task.** Vertical solid lines indicate stimulus onset. Vertical dashed lines indicate flanker onset in the flanker task. The gray area marks the time window of 310 ± 20 ms from target onset. The plots presented rely on raw data and should not be interpreted in terms of main effects, but rather serve to assess the quality of the EEG signal.

**N-back.** The intra-class correlation indicated that differences between subjects explain 15% of the variance. The whole LMM explained 17% of the variance (conditional pseudo $R^2$). It revealed no main effect for cognitive effort investment, no interaction effect for cognitive effort investment × demand, and cognitive effort investment × payoff, all $p > .05$. This also applies to the exploratory analyses yielding no effects on first-order factors (S1 Table).

**Flanker.** The intra-class correlation indicated that differences between subjects explain 17% of the variance. The whole LMM explained 21% of the variance (conditional pseudo $R^2$). It revealed no main effect for cognitive effort investment, no interaction effect for cognitive effort investment × demand, and cognitive effort investment × payoff, all $p > .05$. In line, exploratory analyses yielded no effects on cognitive motivation or effortful self-control (S1 Table).

**Table 8. Results for N2 amplitude analysis in *n*-back and flanker task.**

| Parameter | N-back | | | | Flanker | | | |
|---|---|---|---|---|---|---|---|---|
| | **Beta** | **SE** | ***p*-value** | **Random Effects (SD)** | **Beta** | **SE** | ***p*-value** | **Random Effects (SD)** |
| Intercept | -0.67 | 0.17 | < .001*** | 2.04 | 0.49 | 0.19 | .011* | 2.26 |
| Demand | 0.61 | 0.04 | < .001*** | 0.45 | -0.26 | 0.04 | < .001*** | 0.45 |
| Payoff | 0.01 | 0.02 | .499 | 0.10 | 0.03 | 0.03 | .285 | 0.26 |
| CEI | 0.33 | 0.45 | .460 | | 0.36 | 0.50 | .470 | |
| Demand × Payoff | 0.00 | 0.02 | .897 | 0.15 | 0.01 | 0.02 | .549 | 0.10 |
| CEI × Demand | -0.10 | 0.11 | .375 | | -0.17 | 0.11 | .115 | |
| CEI × Payoff | 0.01 | 0.04 | .728 | | 0.07 | 0.07 | .311 | |
| CEI × Demand × Payoff | -0.04 | 0.05 | .403 | | 0.05 | 0.04 | .218 | |

*Note.* $N_{n\text{-back}} = 147$, $N_{\text{flanker}} = 146$.

***$p < .001$,

**$p < .01$,

*$p < .05$.

CEI = cognitive effort investment.

## P3 amplitude: No changes for individuals with high cognitive effort investment as payoffs increase

Fig 8 shows the grand means of raw ERPs across participants and blocks at P3 relevant electrodes for both conditions and both tasks. Table 9 displays the full results of the LMMs and simple slopes analysis for the significant cognitive effort investment × payoff interaction in the *n*-back task.

**N-back.** The intra-class correlation indicated that differences between subjects explain 13% of the variance. The whole LMM explained 16% of the variance (conditional pseudo $R^2$). It revealed no main effect for cognitive effort investment, and no interaction effect for cognitive effort investment × demand, both $p > .05$. The interaction cognitive effort investment × payoff was significant, $p = .015$ (Fig 9). The simple slopes analysis yielded significant slopes of payoff for low levels of cognitive effort investment, $p < .01$, but not for mean or high, both p > .05. Hence, P3 amplitude only increased in individuals with low cognitive effort investment in response to increasing payoff, whereas the changes were not significant for individuals with medium to high cognitive effort investment. Exploratory analyses revealed only an interaction effect of effortful self-control × payoff, $\beta$ (SE) = -0.06 (0.03), $p = .024$, but not cognitive motivation × payoff, $\beta$ (SE) = -0.05 (0.03), $p = .051$, nor any other effects involving the first-order factors (S1 Table).

**Flanker.** The intra-class correlation indicated that differences between subjects explain 14% of the variance. The whole LMM explained 16% of the variance (conditional pseudo $R^2$). It revealed no main effect for cognitive effort investment, no interaction effect for cognitive effort investment × demand, and cognitive effort investment × payoff, all $p > .05$. This also applies to the exploratory analyses yielding no effects on first-order factors (S1 Table).

## Pupil dilation: No effect of cognitive effort investment

Fig 10 shows the mean pupil size by both conditions for both tasks. Table 10 displays the full results of the LMMs.

**N-back.** The intra-class correlation indicated that differences between subjects explain 8% of the variance. The whole LMM explained 10% of the variance (conditional pseudo $R^2$). It

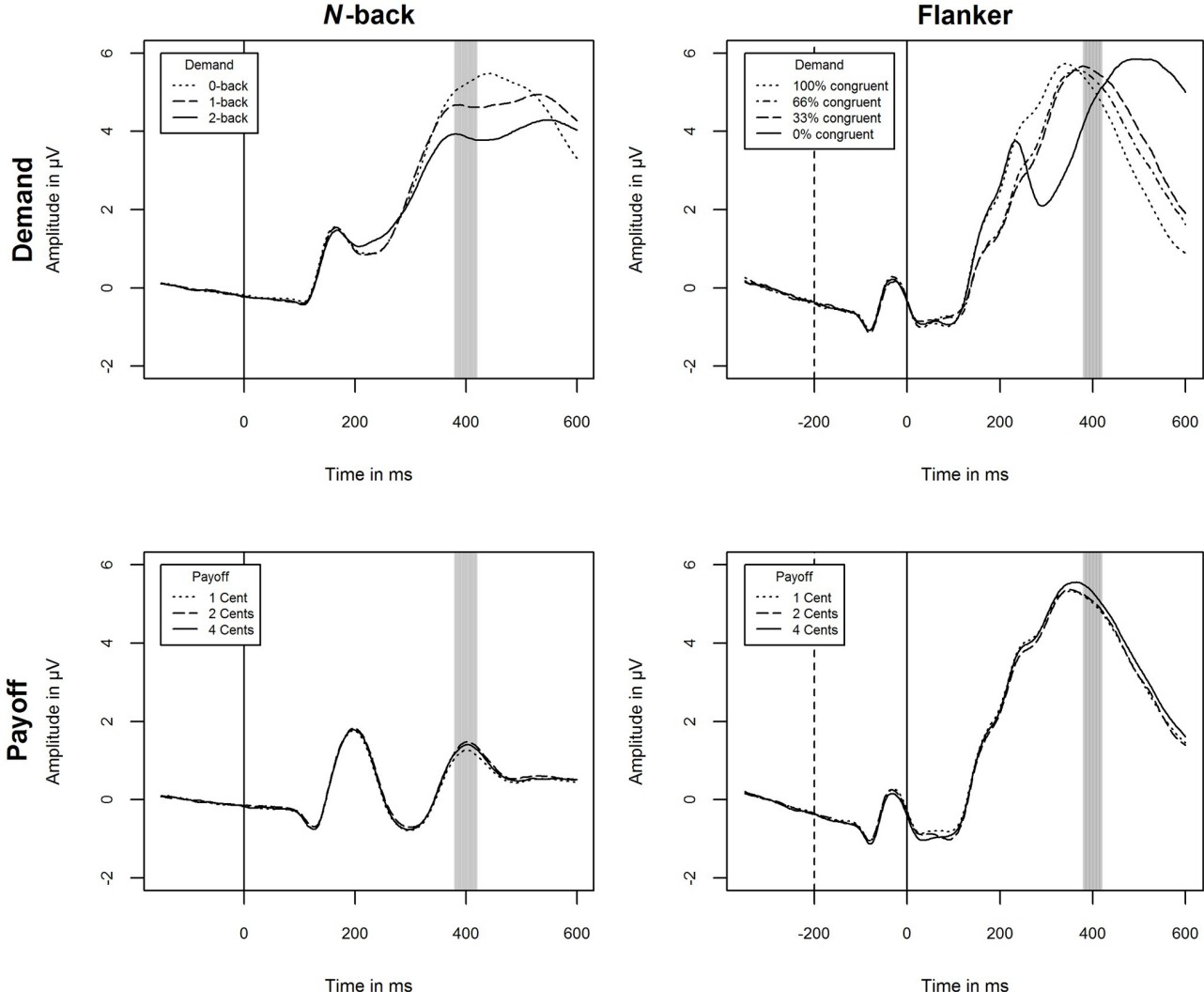

**Fig 8. Mean at ERPs Cz, CPz, and Pz by conditions demand and payoff for the *n*-back and flanker task.** Vertical solid lines indicate stimulus onset. Vertical dashed lines indicate flanker onset in the flanker task. The gray area marks the time window 400 ± 20 ms from target onset. The plots presented rely on raw data and should not be interpreted in terms of main effects, but rather serve to assess the quality of the EEG signal.

revealed no main effect for cognitive effort investment, no interaction effect for cognitive effort investment × demand, and cognitive effort investment × payoff, all $p > .05$. This also applies to the exploratory analyses yielding no effects on first-order factors (S1 Table).

**Flanker.** The intra-class correlation indicated that differences between subjects explain 9% of the variance. The whole LMM explained 15% of the variance (conditional pseudo $R^2$). It revealed no main effect for cognitive effort investment, no interaction effect for cognitive effort investment × demand, and cognitive effort investment × payoff, all $p > .05$. There were also no effects that involved cognitive motivation or effortful self-control in the exploratory models (S1 Table).

## Discussion

Recently, research on *effort* received renewed attention and especially the concept of *effort discounting* has become a topic of interest. In this context, systematic deviations from effort

**Table 9. Results for P3 amplitude analysis in *n*-back and flanker task.**

| Parameter | | N-back | | | | Flanker | | | |
|---|---|---|---|---|---|---|---|---|---|
| | | Beta | SE | p-value | Random Effects (SD) | Beta | SE | p-value | Random Effects (SD) |
| Intercept | | 4.58 | 0.16 | < .001*** | 1.97 | 5.10 | 0.18 | < .001*** | 2.13 |
| Demand | | -0.60 | 0.07 | < .001*** | 0.76 | 0.15 | 0.06 | .011* | 0.64 |
| Payoff | | 0.03 | 0.02 | .121 | 0.16 | 0.08 | 0.02 | < .001*** | 0.19 |
| CEI | | -0.12 | 0.44 | .782 | | 0.08 | 0.47 | .867 | |
| Demand × Payoff | | -0.02 | 0.02 | .406 | 0.14 | -0.01 | 0.01 | .643 | 0.05 |
| CEI × Demand | | -0.24 | 0.17 | .179 | | -0.09 | 0.15 | .554 | |
| CEI × Payoff | | -0.12 | 0.05 | .015* | | 0.01 | 0.06 | .896 | |
| CEI × Demand × Payoff | | -0.03 | 0.05 | .609 | | 0.00 | 0.04 | .978 | |
| | Value of CEI | Beta (SE) | | 95% CI | Conditional Intercept Beta (SE) | | | | |
| Payoff | - 1 SD | 0.08 (0.03)** | | [0.02, 0.13] | 4.63 (0.23) | | | | |
| | Mean | 0.03 (0.02) | | [-0.01, 0.07] | 4.58 (0.16) | | | | |
| | + 1 SD | -0.02 (0.03) | | [-0.07, 0.04] | 4.54 (0.23) | | | | |

*Note.* $N_{n\text{-back}} = 147$, $N_{flanker} = 146$.

***$p < .001$,

**$p < .01$,

*$p < .05$.

CEI = cognitive effort investment.

discounting and demand avoidance behavior were observed and accounted for by individual differences in personality traits such as need for cognition and self-control. These traits are characterized by a heightened motivation to exert effort during goal-pursuit and together form trait cognitive effort investment [8]. In the present study, we were interested in the relationship

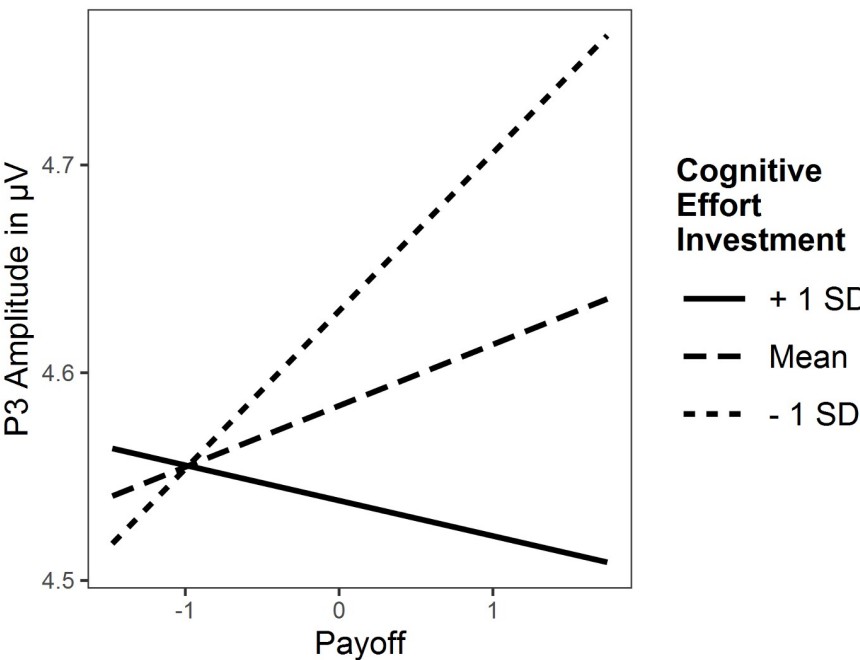

**Fig 9. Interaction plot of cognitive effort investment × payoff interaction for P3 amplitude in the *n*-back task.**

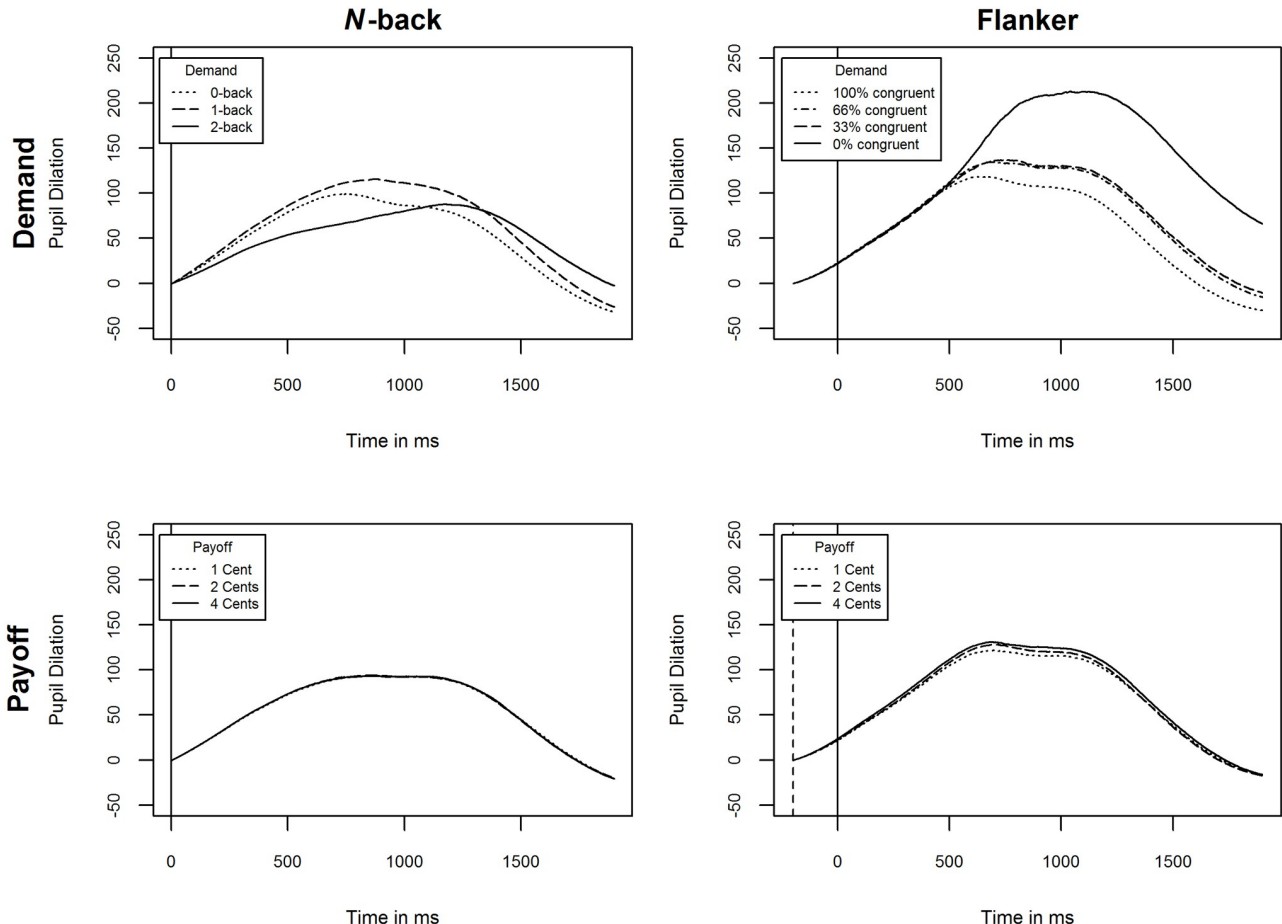

**Fig 10. Mean pupil size by conditions demand and payoff for the *n*-back and flanker task.** Vertical solid lines indicate stimulus onset. Vertical dashed lines indicate flanker onset in the flanker task.

**Table 10. Results for pupil dilation analysis in *n*-back and flanker task.**

| Parameter | N-back | | | | Flanker | | | |
|---|---|---|---|---|---|---|---|---|
| | Beta | SE | *p*-value | Random Effects (SD) | Beta | SE | *p*-value | Random Effects (SD) |
| Intercept | 52.17 | 2.60 | < .001*** | 31.09 | 60.89 | 3.01 | < .001*** | 35.86 |
| Demand | 0.17 | 1.29 | .893 | 14.71 | 8.57 | 0.75 | < .001*** | 7.53 |
| Payoff | 0.15 | 0.35 | .671 | 2.40 | 1.79 | 0.51 | .001** | 4.87 |
| CEI | 1.14 | 6.84 | .868 | | -2.48 | 7.95 | .756 | |
| Demand × Payoff | -0.31 | 0.44 | .492 | 3.37 | 0.34 | 0.34 | .324 | 2.13 |
| CEI × Demand | -1.47 | 3.37 | .663 | | 0.66 | 1.90 | .729 | |
| CEI × Payoff | 0.01 | 0.90 | .988 | | -0.26 | 1.32 | .846 | |
| CEI × Demand × Payoff | -1.27 | 1.16 | .276 | | -0.85 | 0.90 | .345 | |

*Note.* $N_{n\text{-back}} = 146$, $N_{flanker} = 144$.

***$p < .001$,

**$p < .01$,

*$p < .05$.

CEI = cognitive effort investment.

between trait cognitive effort investment and actual effort investment during task processing. We focused on the main questions: (1) Is cognitive effort investment related to objective effort indices? (2) Does cognitive effort investment moderate the relation between effort indices and demand and payoff? To this end, we measured subjective load, reaction time, accuracy, (early and late) FMθ, N2 amplitude, P3 amplitude, and pupil dilation during the performance of two typical cognitive control tasks, i.e. *n*-back and flanker task, with varying demand and payoff. Our findings provide the following picture.

The mixed effects models revealed a main effect of cognitive effort investment for accuracy in the *n*-back task, indicating that individuals scoring high on cognitive effort investment generally perform better than individuals scoring low. Looking now exploratory at the subdimensions, the main effect is still true for effortful self-control, but not cognitive motivation. This suggests that individuals perform more accurately due to control rather than motivational processes. Moreover, effortful self-control is a reliable predictor of perceived task load in the *n*-back task. Hence, individuals with high levels of effortful self-control generally experience lower effort during task processing. This is consistent with the meta-analytic findings of de Ridder et al. [12] that self-control correlates more strongly with automatic than controlled behavior, suggesting that individuals with high self-control tend to establish automatisms being unconscious and less effortful. Moreover, it strengthens our assumption on individuals scoring high on cognitive effort investment to generally invest effort more efficiently. Thus, cognitive effort investment reliably predicts accuracy in the n-back task, but not any other effort index (H1).

We also found no correlation between perceived task load in the cognitive task battery or the flanker task and cognitive effort investment. First-order factors were also not correlated. This suggests that cognitive effort investment does not negatively correlate with perceived task load (H2), i.e. individuals perceived task load equally and independently of their level in cognitive effort investment.

Furthermore, there was an interaction effect of cognitive effort investment and demand for early FMθ power in the flanker task. Although all individuals–regardless of trait level–dynamically allocated more cognitive resources according to task demands, the increase steepened as levels of cognitive effort investment increased. This is consistent with the predictions of Cacioppo [11] that cognitive effort increases more for individuals with high compared to low need for cognition as demands increase. Moreover, it reflects the results of Mussel et al. [13] for late FMθ power in the *n*-back task, even if we were not able to replicate this specific effect. In this context, a descriptive examination shows that late FMθ power was generally lower for individuals with high compared to low cognitive effort investment in both tasks, just as in the aforementioned study. The exploratory analyses of early FMθ power revealed an interaction effect for effortful self-control, but not cognitive motivation suggesting that the effects are more attributable to control than motivational processes. Hence, early FMθ power showed a stronger change in individuals with high (vs. low) cognitive effort investment with increasing demand in the flanker task, but not all other indices of cognitive effort in either tasks (H3).

The mixed effects models yielded an interaction effect of cognitive effort investment and payoff for reaction time in both tasks as well as P3 amplitude in the *n*-back task. The simple slopes analyses for reaction time revealed the predicted pattern: Individuals with low levels of cognitive effort investment respond faster, i.e. exert more effort, whereas individuals with high levels show slower reaction times, i.e. exert less effort, as a response to increasing payoff. This mirrors the observations of Sandra and Otto [14] and meets our predictions that individuals with low levels of cognitive effort investment are strongly motivated by external incentives, whereas this is not the case for individuals with high levels due to their intrinsic motivation. Additionally, exploratory models of both first-order factors showed this effect indicating that

motivational and control processes play an equal role in this context. The results on the P3 amplitude in the *n*-back task show a neural correlate of this effect. Simple slope analyses showed that P3 amplitude only increased in individuals with low cognitive effort investment in response to increasing payoff, whereas the changes were not significant for individuals with medium to high cognitive effort investment. This suggests that low compared to high levels in cognitive effort investment are associated with a stronger valuation of the amount of payoff. The exploratory analyses of P3 amplitude revealed an interaction effect for effortful self-control, but not cognitive motivation which implies that this effect is more strongly attributed to control than motivational processes. The *self-determination theory* of Deci and Ryan [65] provides a larger theoretical framework for our observations. The theory predicts positive effects of incentives for externally motivated individuals (corresponding to low levels of cognitive effort investment) and no or even detrimental effects for intrinsically motivated individuals (corresponding to high levels of cognitive effort investment) as they undermine intrinsic motivation. Taken together, reaction time and P3 amplitude change less in individuals with high (vs. low) cognitive effort investment with increasing payoff, whereas other effort indices may not (H4).

The conclusion for our two main questions is (1) that cognitive effort investment is related to accuracy (*n*-back), but not to other objective effort indices and (2) that it moderates the relation between early FMθ power (flanker) and demand, but not other effort indices, and between reaction time (*n*-back and flanker) as well as P3 amplitude (*n*-back) and payoff in typical cognitive control tasks. Taken together, our results partly support the notion that individuals with high levels of cognitive effort investment exert effort more efficiently. Moreover, the notion that these individuals exert effort regardless of payoff is partly supported, too.

In the following, we discuss these results from four perspectives. First, we have a closer look at the sensitivity of the effort indices used. Second, we briefly discuss the role of cognitive abilities and crystallized intelligence. Third, we take the conservative view, i.e. trait effects actually exist, but we were not able to detect them fully. Fourth, we take the pessimistic view, i.e. we did not detect trait effects fully because they do not exist.

## Cognitive effort markers are sensitive to demand manipulations but hardly to payoff

We used a multi-dimensional approach to measure effort exertion by combining subjective, behavioral, electrocortical and psychophysiological indices. However, we only found partial evidence for trait effects during the performance of two different cognitive control tasks with varying demand and payoff. The question arises whether trait effects did not emerge fully due to an unsuccessful manipulation, inappropriate markers used or an inadequately assumed linear relationship between effort investment and task demands.

Cognitive effort markers are sensitive to demand manipulations, but partially show task-specific patterns. All indices (except for late theta and pupil dilation in the *n*-back task) produced significant main effects of demand in both tasks. However, a closer look at the effects reveal opposite directions for the N2 amplitude in the *n*-back task as well as early and late FMθ, and P3 amplitude in the flanker task. The opposite patterns in electrocortical activity may indicate that different demands generate different forms of effort, i.e. updating (*n*-back) vs. inhibition (flanker). Even performance measures in different inhibitory control tasks correlate to a lower extent than theoretically expected [66], i.e. $.10 < r_{RT} < .61$ and $.02 < r_{ER} < .29$. Additionally, task requirements affect the temporal dynamics of FMθ [67]: proactive control leads to a focal activation at frontal sites, whereas reactive control leads to a broader activation. Moreover, Nigbur et al. [30] observed that cognitive inhibition tasks produce different interference effects as mirrored by FMθ and N2 amplitude. The largest effect in FMθ occurred in the NoGo task,

followed by the flanker task and the Simon task. The N2 amplitude in the interference condition was largest in the NoGo task, followed by the Simon task and the flanker task. Taken together, the brain seems to respond in a more complex way than to produce homogeneous stimulus–response patterns in every effort marker across a variety of cognitively demanding situations. Moreover, there may be crucial differences in a trial-wise (flanker) vs. a block-wise (*n*-back) manipulation of demand. Future studies should explore this issue in more detail.

Cognitive effort markers are partly sensitive to payoff manipulations and yield task-specific sensitivity. Reaction time, P3 amplitude and pupil size are sensitive effort markers that significantly indicated increasing payoff in the flanker task, but not in the *n*-back task. Bonner et al. [68] reviewed 131 published laboratory studies regarding the effect of payoff on task performance. Indeed, payoff increased performance only in 30% (worst-case scenario) to 50% (best-case scenario) of the experiments. In the remaining experiments, payoff had no or even a negative effect on performance. The authors explain these findings with the complexity of tasks that increases the gap between demand and skill: the higher the complexity, the lower the likelihood of performance improvement by payoff. Our findings mirror this complicated relation between payoff and effort outlined in the review, demonstrating that increasing payoff may not necessarily elicit the same effort in different tasks.

We generally assumed that actual effort investment is linearly related to demand and payoff (cf. Fig 2). This assumption results from previous findings that suggest linearity [6, 13, 21, 22]. However, an inverted U-shape relation is theoretically conceivable, if task demands exceed cognitive abilities. Indeed, we observed high frustration and low commitment in participants during *n*-back levels with $n > 2$ in our pilot study. Therefore, we implemented *n*-back levels with $n = 0, 1, 2$ instead of $n = 1–6$ like in the underlying study by Westbrook et al. [6].

## Cognitive abilities and crystallized intelligence do not alternatively predict results

Differences in effort indices cannot be better explained by cognitive abilities or crystallized intelligence. Considering the implementation of two typical cognitive control tasks and the relatively large correlation of the indicator variables of cognitive effort investment with intelligence and school grades [9, 15–17], an alternative explanation of the presented results by cognitive abilities and crystallized intelligence is conceivable. In order to control for a possible influence and to ensure that the results are reliable, we added both into our statistical models as additional predictors. This has proven to be reasonable for some of the effort indices expressed by significant main effects. As a conclusion, results presented exist over and above cognitive abilities and crystallized intelligence.

## The conservative view of failed detection of (consistent) trait effects

From the conservative point of view, we may have not been able to detect the predicted trait effects fully even though they actually exist. Therefore, we critically evaluate the limitations of the present study. We identified three potential sources that could have influenced the results. These are the size and homogeneity of our sample, the choice of typical cognitive control tasks and the researcher's degrees of freedom.

With regard to sample size, we were able to find medium-sized effects with reasonable power and at a corrected alpha-level. Yet, as detailed below, the assumption of medium-sized effects in individual differences research might be overly optimistic. In addition, the homogeneity of our sample resulted in too little trait variance. Our study sample consisted of a highly educated (97% holding a matriculation standard), student sample (86% students) with a high motivation regarding the study, i.e. they attended twice and performed a series of demanding

tasks. Indeed, intelligence shows medium correlations to need for cognition and intellect (r > .20) [9, 41, 69]. Thus, the characteristics of our sample may have led to a ceiling effect and have not left enough variance in these traits and, therefore, in cognitive effort investment. We propose a replication of the present study with a sample that represents a broader population with more variance in the traits of interest.

Typical cognitive control tasks produce strong situations that suppress the expression of individual differences. The interplay between individual predispositions and the context for action is called *person-situation interaction*. Mischel [70] assumed that situations leave varying degrees of freedom to express individual differences. Thus, *weak situations* provide a maximum of degrees of freedom, whereas *strong situations* provide a minimum. In line, Chevalier [71] found a significant correlation between subjective values for 1 vs. 2-back choices and need for cognition ($r = .37$) in an adapted COG-ED task, but not for 1 vs. 3-back choices ($r = .09$). The trait effect disappeared in the hard demand condition producing a strong situation. As typical cognitive control tasks were developed to find strong general behavioral effects, they produce strong situations with no room for trait variance. Gärtner et al. [18] examined the relation of need for cognition and a variety of cognitive control tasks, but found no correlation between need for cognition and performance in any of these tasks. The development of appropriate tasks for personality research should be the next step in further research.

The researcher's degrees of freedom enhance the probability of false-positive results [72]. We have carefully chosen our tasks, cognitive effort indices, preprocessing and analysis pipelines basing on previous literature. However, all these decisions may have had a substantial influence on results. In the preceding paragraph, we discussed the *n*-back and flanker task as typical cognitive control tasks in terms of producing strong situations. We examined a variety of cognitive effort indices. However, physiological reactions are quite complex. It is conceivable that these reactions are as individual as behavioral responses and that it is the pattern of responses that explains individual differences rather than individual indicators. As to the preprocessing, we applied fixed time windows to average the N2 and P3 amplitude and fixed frequency bands to detect FMθ for all subjects basing on objective criteria. However, latencies of amplitudes as well as frequency bands vary between individuals, e.g. alpha frequency range varies depending on age, neurological diseases, memory performance, brain volume and even on task demands [24]. Thus, in the worst case the alpha and theta band may overlap and cancel each other out. Klimesch [24] recommends an individual over objective determination of latencies and frequency bands as well as an adjustment for recording site. Individual definitions are only one out of a whole series of preprocessing decisions in the garden of forking path. Šoškić et al. [73] reviewed the preprocessing pipelines for the picture-evoked N400 and found a huge inconsistency in methodological decisions. The multiverse analysis in the follow-up study illustrates the influence of different decisions on results [74]. At the moment, researchers do best to disclose these influences via multiverse analyses or to reduce this flexibility by consequently following *a priori* decisions basing on previous literature, whereas the noise in e.g. EEG-recording is better processed data-driven.

## The pessimistic view of nonexistence of (consistent) trait effects

A conclusion from our findings could be a pessimistic one: we could not find trait effect fully because they do not exist. A meta-analysis by Gignac and Szodorai [75] systematically examined the distribution of correlation magnitudes in individual differences research and found that effect sizes typically fall into the $.20 < r < .30$ range. This fact could to some extent be countered by a larger sample which would also have a positive effect on the reliability of findings as the likelihood of false-positive results declines with increasing sample size [72].

However, a large sample size is no guarantee for a successful replication. The Many Labs Replication Project put effort in replicating 100 psychological studies in order to estimate reproducibility [76]. In summary, the collaboration reproduced only one third of originally significant results. If effects replicated, the magnitude of the mean effect size was only half the originally reported size [76]. Indeed, there exist only few replications of the four seminal studies on traits need for cognition and self-control, i.e. Kool et al. [4], Westbrook et al. [6], Mussel et al. [13] and Sandra and Otto [14] with quite lower effects. Kramer et al. [20] replicated the relation of need for cognition and effort discounting of Westbrook et al. [6], but required a far larger sample size ($N = 294$) to observe a smaller effect size ($r = .13$ as compared to $r = .32$). Yet, the effect disappeared when controlling for $n$-back performance in exploratory analysis. With a sample size of $N = 217$, Strobel et al. [45] attempted to conceptually replicate the findings of Kool et al. [4] on the relation of self-control and demand avoidance. Despite the fact that both behavioral demand avoidance and cognitive effort investment were trait-like, they did not significantly intercorrelate ($r = -.08$). A registered report by Crawford et al. [77] seems very promising in this context aiming at a sample size of $N = 300$. However, the pilot data with $N = 31$ yielded no relation between need for cognition and effort discounting in the COG-ED task. Taken together, relations between the actual investment or avoidance of cognitive effort and dispositional differences in the willingness to invest mental effort seem to be small at best and might–if at all–only be observable in very large samples. This raises the question of the practical relevance of such effects.

## Conclusion—Implication for research

The aim of the present study was to further investigate the relationship between trait cognitive effort investment and actual effort investment during task processing. We assumed (1) that cognitive effort investment is related to objective effort indices and (2) that it moderates the relation between effort indices and demand or payoff, respectively.

Taken together, our results partly support the notion that individuals with high levels of cognitive effort investment exert effort more efficiently. Moreover, the notion that these individuals exert effort regardless of payoff is partly supported, too. Future research on individual differences should consider large-scale replications in order to detect trait differences and to produce reliable results. This is also desirable for further replications of the findings of Kool et al. [4] and Westbrook et al. [6] to clarify the influence of need for cognition and self-control on demand avoidance and effort discounting. Additionally, researchers interested in cognitive effort investment should select their sample from a broader range in the population to avoid restriction of variance in the traits of interest. Yet, a much harder task for personality research we see in the development of appropriate executive control tasks that produce weak situations in which individual differences can unfold. Taken together, the present study may further our understanding of the conditions under which person-situation interactions occur, i.e. the conditions under which situations determine effort investment in goal-directed behavior more than personality, and vice versa.

## Supporting information

**S1 Table. Results for LMM and GLMM analyses on first-order factors for $n$-back and flanker task.** *Note.* $N \geq 144$. ***$p < .001$, **$p < .01$, *$p < .05$. COM = cognitive motivation, ESC = effortful self-control.
(DOCX)

**S2 Table. Simple slopes analysis results for all effort indices in *n*-back and flanker task.** *Note.* $N \geq 144$. $^{***}p < .001$, $^{**}p < .01$, $^{*}p < .05$. CEI = cognitive effort investment. (DOCX)

**S1 Fig. Effect sizes depending on power in the *n*-back and flanker task.** $J = N$ = sample size, $n$ = trials per condition, $\rho$ = intra-class correlation. (TIF)

**S2 Fig. Johnson-Neyman plots for all effort indices in *n*-back and flanker task.** (TIF)

**S3 Fig. Plots of interaction between cognitive effort investment × demand or payoff in *n*-back and flanker task for all effort indices.** (TIF)

## Acknowledgments

The authors thank Heike Buchantschenko, Vincent Kirchner, Viviane Lange, Patricia Schimm, Pia Wilhelm, Josephine Zerna, and Madeleine Zoglauer for their assistance in data collection. We thank Gesine Wieder for her assistance in data collection and in project coordination during parental leave. We thank Maarten Lars Jung and Johannes Rodrigues for the valuable exchange on linear mixed models and sample size calculation.

## Author Contributions

**Conceptualization:** Corinna Kührt, Philipp C. Paulus, Alexander Strobel.

**Data curation:** Corinna Kührt, Sven-Thomas Graupner.

**Formal analysis:** Corinna Kührt, Alexander Strobel.

**Funding acquisition:** Alexander Strobel.

**Investigation:** Corinna Kührt, Sven-Thomas Graupner.

**Methodology:** Corinna Kührt.

**Project administration:** Corinna Kührt.

**Resources:** Corinna Kührt.

**Software:** Corinna Kührt, Sven-Thomas Graupner, Philipp C. Paulus.

**Supervision:** Corinna Kührt, Sven-Thomas Graupner, Alexander Strobel.

**Validation:** Corinna Kührt, Sven-Thomas Graupner, Philipp C. Paulus, Alexander Strobel.

**Visualization:** Corinna Kührt, Alexander Strobel.

**Writing – original draft:** Corinna Kührt, Sven-Thomas Graupner, Philipp C. Paulus.

**Writing – review & editing:** Corinna Kührt, Sven-Thomas Graupner, Philipp C. Paulus, Alexander Strobel.

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
