## [Decision Letter · Decision Letter 0]

13 Mar 2023

PONE-D-23-01823Cognitive effort investment: Does disposition become action?PLOS ONE

Dear Dr. Kührt,

Thank you for submitting your manuscript to PLOS ONE. After careful consideration, we feel that it has merit but does not fully meet PLOS ONE’s publication criteria as it currently stands. Therefore, we invite you to submit a revised version of the manuscript that addresses the points raised during the review process.

The reviewers are generally supportive and find your manuscript well written. In agreement with the reviewers, I also find the manuscript interesting, well-developed, and its topic in line with the scope of the journal. However, both reviewers had concerns and raised a number of questions regarding both the methods (see mainly Reviewer 1) and the theoretical rationale (see mainly Reviewer 2) of the study. The concerns emerged by the Reviewers require changes to the manuscript on a number of points.

We look forward to receiving your revised manuscript.

Kind regards,

Árpád Csathó, Ph.D.

Academic Editor

PLOS ONE

Journal Requirements:

Additional Editor Comments :

Dear Dr. Kührt,

hereby, I would like to inform you that we have received the reviews about your manuscript from two expert reviewers. The reviewers are generally supportive and find your manuscript well written. In agreement with the reviewers, I also find the manuscript interesting, well-developed, and its topic in line with the scope of the journal. However, both reviewers had concerns and raised a number of questions regarding both the methods (see mainly Reviewer 1) and the theoretical rationale (see mainly Reviewer 2) of the study. The concerns emerged by the Reviewers may require changes to the manuscript on a number of points, but please prepare the revised version of your manuscript following each of the reviewers' comments and suggestion.

Thank you for your submission,

sincerely Yours,

Arpad Csatho

Reviewers' comments:

Reviewer's Responses to Questions

**Comments to the Author**

1. Is the manuscript technically sound, and do the data support the conclusions?

Reviewer #1: Yes

Reviewer #2: Partly

2. Has the statistical analysis been performed appropriately and rigorously? 

Reviewer #1: Yes

Reviewer #2: Yes

3. Have the authors made all data underlying the findings in their manuscript fully available?

Reviewer #1: Yes

Reviewer #2: Yes

4. Is the manuscript presented in an intelligible fashion and written in standard English?

Reviewer #1: Yes

Reviewer #2: Yes

5. Review Comments to the Author

Reviewer #1: Thank you very much for the opportunity to review this manuscript, which I did with great pleasure. Indeed, there is a lot to like about this study: Hypotheses are grounded soundly in the literature; the design is well-planed; there are multiple tasks to generalize across task-specific effects; the sample size is relatively large; there are multiple indicators of effort investment, including subjective load, reaction time, accuracy, theta power, ERPs, and pupil dilation, and the analyses are sound. As none of my comments is "overly major", I present them in the order in which they appeared in the manuscript.

In the abstract and the introduction (but not later on), the authors use the concept of boredom to explain why individuals with high compared to low levels of cognitive effort investment deliberately chose harder tasks or increase their effort on demanding tasks. I doubt that the latter effect is caused by boredom avoidance. First, the absence of boredom is not necessarily cognitively demanding activity (think of simple leisure activities to kill time, such as watching television, needlework, or sitting by the fire). Second, from the early work of Berlyne to Zuckerman's sensation seeking, boredom susceptibility has been found to be a distinct dimension (e.g., from specific curiosity or experience seeking). To me, it seems more likely to consider cognitive effort investment as an approach motivation (directed towards the goal of mastering an intellectual challenge), rather than an avoidance motivation (i.e., avoiding boredom).

Please give more information about how the sensitivity analysis was performed, at the very least by giving a citation - there are quite a few different ways to do so (for example, it seems that number of trials per condition was considered).

For the cognitive effort investment measures, please give some more information about the CFA (line 239). Which variables were used as indicators (items, parcels, scales, overall test scores)? Regarding the intellect scale, the sub-dimension of conquer seems rather an indicator of effortful self-control than cognitive motivation. Also, I was a bit worried that the two indicators of effortful self-control (self-control and effortful control) had no reference to epistemic behavior, but still were used as indicators of cognitive effort investment. The example item of the effort control scale is about sitting still while energized, which seems unrelated (or even the opposite) of the tendency to engage in cognitively challenging activities.

Related to the above, I would be interested to see the effects for the two subdimensions (effortful self-control and cognitive motivation). I understand that this produces a whole lot of additional tests. Describing them as additional exploratory analyses for future studies or meta-analyses (without the need for correcting for alpha inflation) or briefly summarizing any differences on the level of these dimensions, compared to the overall score (cognitive effort investment) would be a valuable additional value to the literature.

Regarding the NASA-TLX, did the principal component analysis support the computation of a one-factor solution (line 245)? Subjective perception of mental demands seems very close to what the authors wish to measure. on the other hand, subjective perceptions of physical and temporal demands seem much less relevant for the task at hand. Finally, subjective perception of performance and effort may even be negatively related to perceive task load (the easier the task, the better the performance). Thus, are factor scores across all six indicators the best way to assess what you aim to measure?

Regarding the description of the n-back task (line 289), the authors first note four levels of demands (0-back, 1-back, 2-back, 3-back), but later on only three levels (level (0, 1, 2), line 304).

Regarding the Flanker tasks, the author mention six blocks, but only three seem necessary for manipulating payoff (0.01, 0.02, 0.04 EUR). Why are there six blocks?

Any reason why the Gratton & Coles procedure to remove blink artefacts is used, rather than more recent approaches like ICA (see 10.3389/fnins.2021.660449).

Please specify how the windows for the ERPs were specified (peak detection on the grand mean?). Why was such a small window chosen for the large P3-component?

Line 439: The authors note that they centered all level 1 predictors within person. If I understood correctly then level 1 predictors are "demand" and "payoff", i.e., experimentally controlled variables which should have equal means across participants anyway. Therefore, why center within persons?

This might be a matter of debate, but I wondered whether corrections for multiple testing were overly strict. If I understood correctly, results are corrected family wise for seven independent tests according to the seven indicators of cognitive effort (i.e., reaction time, accuracy, early and late theta power, N2 and P3, and pupil dilation). Corrections for multiple testing are appropriate whenever multiple tests are performed to test one substantial hypothesis. As the hypotheses are formulated for "effort indices", the authors' approach is generally appropriate. It implies that a hypothesis is confirmed whenever one of these tests is significant on the adjusted significance level. However, from a more substantively point of view, this sems very strict. The approach corresponds with the assumption of response coherence, i.e., effects of personality traits are equivalent across responses (behavioral, psychophysiological, etc.). It is well known that this is not the case as effects are strongly depending on the respective response (e.g., Lang, 1971; Asendorpf, 1988). Thus, one could also refrain from corrections for multiple testing, while interpreting a significant effect as confirming the hypothesis for this particular response, rather than cognitive effort in general (i.e., across indices).

According to the (selective) results presented for levels of demand on responses in the Flanker task (see Figures 7 and 8), I wondered whether it is appropriate to use "demand" as a continuous predictor. Levels of demands (especially "0%") seem to have qualitatively different effects. This suggests that a categorical variable might be more appropriate.

The discussion is mainly about why the expected effects were not found. Maybe related to the above issue of corrections for multiple testing, I found this perspective to be one-sided and very strict. The interaction effects between CEI and payoff for reaction times are well in line with Deci and Ryan's self-determination theory which predicts positive effects of incentives to have positive effects for externally motivated individuals (which corresponds to low levels of CEI) and no or even detrimental effects for intrinsically motivated individuals (corresponding to high levels of CEI) as they undermine intrinsic motivation. Results for P3-amplitudes (Fig. 9) show a neural correlate of this effect. Additionally, the significant interaction between CEI and demands in the Flanker task on early theta is well in line with predictions by Cacioppo who predicted stronger increases in cognitive effort for high compared to low NFC individuals as demands increase, and mirrors results from Mussel et al. for late theta. Also, there was a significant effect of CEI on accuracy in the n-back task that was not further interpreted as not being robust when controlling for multiple testing. In my opinion, these results merit to be acknowledged.

When discussing differential effects for n-back and Flanker task (e.g. page 39), you may also mention block-wise vs. trial-wise manipulation of demands in the two tasks which may have profound effects on the results (e.g., https://doi.org/10.1093/scan/nsab126).

Check whether you can avoid some abbreviations that are rarely used to reduce the cognitive load for (especially low CEI-) readers, e.g., SSA, ICC, FDR.

Reviewer #2: **** SUMMARY & GENERAL COMMENTS: ****

This study investigates to what extent dispositional scores on a newly composed cognitive effort investment scale are related to a range of behavioral and psychophysiological markers of mental effort and whether they moderate the effect of task demand and payoff on mental effort exertion.

This is a thorough and interesting study with well documented methods, including the openly available data and analyses scripts. My main concerns regard clarification (and justification) of the specific predictions and operationalization of mental effort with the different measures used in this study. Furthermore, while I appreciate the multidimensional approach, it makes it somewhat difficult to synthesize the findings. I feel that the authors’ motivation to do this, their predictions regarding the specific measures, and their expectations regarding the overall picture of findings (across measures) could be specified in a bit more detail in the introduction.

Detailed comments on this and other issues are outlined below.

**** MAJOR COMMENTS: ****

(1) The predictions regarding moderation of demand and payoff effects on effort by interindividual differences in the cognitive effort investment trait component (CEI) need more explanation and justification (particularly p.3-4 & Fig. 2).

(1.a) Regarding demand:

- If individuals scoring high on CEI increase their actual effort investment more with increasing task demands than those scoring low, this may be due to a number of reasons. They may have more cognitive resources available, and/or be better at detecting demand-differences and allocate effort accordingly, and/or be more motivated to try and thus not give up as easily as those scoring low, etc. Maybe the authors could specify what they assume might underlie such a moderation effect; especially given their CEI component includes aspects of both motivation and self-control.

- The general relationship between task demands and actual effort investment may not be linear (as presented here, Fig.2) but follow an inverted U-shape, where, as demands exceed resources, detachment from the task at hand occurs and effort decreases (see e.g. van Steenbergen et al., 2015, doi: https://doi.org/10.3389/fpsyg.2015.00974). In theory, it could then be expected that interindividual differences are not necessarily in slope, but in ‘turning point’ (peak of the inverted U-shape). To what extent such a non-linear pattern emerges may depend on the task at hand and the different demand levels included, and might thus not be expected for the tasks used in this study. Nevertheless, the authors may want to consider this and mention it briefly when presenting their linear theoretical model.

(1.b) Regarding payoff:

- Particularly the prediction that high CEI scorers would increase actual effort less with increasing payoff needs more justification. Why would they care less about higher payoffs than lower scoring individuals? Again, multiple interpretations are possible, and I hope the authors can clarify their standpoint on this. It may be that high scorers don’t have to increase their effort with payoff because they are ‘already giving it all’ at low payoffs. This may also be demand dependent: at high demands, it might be more likely for low scorers not to increase their effort investment with increasing payoffs, as it may not be worth it (‘giving up’), whereas high scorers may still be ‘up for the challenge’ - this would result in opposite effects than those predicted here. Another possibility, and perhaps closest to the authors’ interpretation, is that low and high scorers differ in their motivation to participate in the study to begin with. While low scorers may be motivated by the potential rewards, high scorers may participate mostly because they enjoy such tasks.

- Note here also a lack of clarity regarding the following formulation (p.4): ‘They do not benefit from payoff in terms of improved performance or recruiting more resources in order to improve performance.’ – What do the authors mean by ‘benefit’?

(1.a+b) The supporting evidence cited for both predictions (p.4, l.64-70) should also be explained with a bit more detail:

- The Mussel, Ulrich [12] study found that individuals with high levels of need for cognition allocate their cognitive resources according to task demands. But what about those with low levels? Only if they differ would this support the above-mentioned prediction.

- The Sandra and Otto paper [13] should also be discussed more carefully, as it (unexpectedly) shows not a lower adaptation of effort to payoff in those with high need for cognition, but a negative one. This is not in line with predictions by the authors of the current study.

(2) The role of cognitive ability:

- In several sections of the paper, it is stated that individuals with high levels of dispositional CEI are ‘under-stimulated by easy tasks’, ‘find task demands easier’, ‘exert effort more efficiently’, etc. This suggests a potential link to cognitive ability/resources, which is only briefly (and comparatively late; p.7, l.141-151) mentioned in the introduction where it stands somewhat in isolation. In contrast, the statements mentioned above seem to be scattered somewhat unsystematically throughout the paper, not directly linked to the question of cognitive ability/resources, and it is a bit unclear what the authors meant by this and (in part) how they expect this would translate into findings of their study (especially regarding ‘exert effort more efficiently’) .

I think the question of whether effort investment differences may be related to differences in cognitive ability/resources should be addressed early and more systematically in the introduction, and it should be made clear whether statements such as ‘under-stimulation’ or ‘efficient effort allocation’ are meant to be related to this.

(3) Accuracy, reaction times, and allocating effort ‘efficiently’:

- There is a difference between investing effort per se and investing effort efficiently. This paper seems to refer to one concept in some and the other in other sections. It needs to be made clearer what the focus is, whether interindividual differences are expected for both or just one of the constructs, and how this would be reflected in the different effort indices.

- When introducing the different measures of effort, most of the evidence provided to justify them as ‘effort measures’ regards their association with task load/demand. Naturally, effort should be higher under higher demands, i.e. it is reasonable to assume that these measures also reflect invested effort. However, especially for the behavioral measures, that is not necessarily the case. If accuracy is low, one cannot know whether not enough effort was exerted, or whether all the effort was exerted, but not successfully. Regarding reaction times (RTs), the presented picture is even more unclear. They are first presented as ‘longer under high task load’, which suggests that more effort should translate into longer RTs. But in their own study, the authors take shorter reaction times to reflect ‘successfully invested effort’, and longer to reflect lower effort (similar to the cited Sandra and Otto study [13]).

(4) Results:

- Make clear which (and how) analyses were corrected for cognitive abilities and crystallized intelligence and report the associated findings (e.g. main effects).

(5) Methods:

- p.17, linear mixed effects model: Why does the model include a 3-way interaction when this is not a question the authors set out to answer? I do believe this interaction is relevant to fully understand the interplay between demand- and payoff-motivated effort investment, and the moderation thereof by interindividual differences/preferences. However, given this question was not one the authors addressed in their introduction, its inclusion here is a bit surprising.

I would suggest either to update introduction and predictions accordingly, or to restrict the model to those 2-way interactions actually relevant to the hypotheses.

(6) Discussion:

- p.35, l.659-662: It is unclear that (and why) the here presented RT results were predicted (see also general comment above).

The RT-findings should be interpreted carefully and discussed more thoroughly.

Note that higher demands lead to longer RTs in both tasks, and higher payoff to longer RTs in the nback- but shorter in the flanker-task. This suggests a different association of RTs with effort in the different tasks. Given the flanker-task is more fast paced (response time limit = 1200ms), maybe the speed-accuracy trade-off is more prominent here? I.e., one may ‘risk’ getting it wrong after shorter deliberation and focus on ensuring one responds in time when pay-offs are high.

- p.40, ‘Taken together, our results do not support the notion that individuals with high levels of cognitive effort investment exert effort more efficiently.’

It is unclear that and why this was expected and how this ‘efficiency’ was operationalized with the different effort indices (see general comment above).

**** MINOR COMMENTS: ****

(7) The term ‘effort discounting’ is used a bit imprecisely throughout the manuscript (e.g. p.3, l.28-29). Effort discounting typically describes the principle that the subjective value of a reward decreases as the amount of effort required to obtain it increases (e.g. p.16 in [2], i.e. Botvinick et al., 2009). Instead, the authors here seem to use the term interchangeably with the ‘law of least mental effort’.

(8) Introduction:

- p.5, l.100-102: Explain that switch costs here reflect RTs and that here (unlike in the second sentence of the same paragraph) one assumes shorter RTs reflect more (efficiently allocated) effort (see also major comments on RTs).

- p.7, (H2): This hypothesis has not been well prepared/justified in the introduction. The authors should add a short explanation on why they assume perceived task load is lower in high CEI scorers (see also major comments on CEI-based predictions and role of cognitive abilities).

(9) Methods:

- ‘Participants completed 20 training trials for each difficulty level until they completed at least 85 % trials correctly.’:

Does that mean 20 training trials per difficulty level were enough for all participants to reach the 85% correct criterion, or were sets of 20 trials per difficulty level repeated until this criterion was reached? If the latter: consider providing total number of required training trials per task and testing whether this was associated with CEI.

- It seems blocks differed by demand in the n-back but by payoff in the Flanker task? This block design could be explained a bit more clearly in the descriptions of the tasks, and prior to stating things like ‘at the end of each block’.

- Pupil size preprocessing: if I understood correctly, all eye-blinks and other artifacts were treated as missing, and not interpolated. This may lead to a lot of data loss, but it also makes the remaining pupil size measures somewhat more reliable (reflecting the ‘real’ pupil size), so I don’t argue against this approach per se. However, I wonder how much data was lost overall. When the authors state that ‘Trials with missing data, set to NA or with RTs < 100 ms were excluded from statistical analysis’ (p. 17), does that mean that pupil size averages for the time windows of interest were not calculated if any samples within the respective time window were missing? It would be good to report how many trials after data exclusion were available per participant and per task in the end.

- p.17, l. 406-407: I believe the reference here should be [56], not [29]

(10) Results:

- Use decimal points (not comma) in all presented numbers (see tables)

- p.19, ‘Manipulation check: Effort indices reflect levels of demand and partly of payoff’:

Maybe state explicitly that the p-values presented here refer to the main effects presented in the tables 2-9. I would also add a short note on the direction of the effects.

- p.20, ‘Perceived task load: No relation to cognitive effort investment’:

Why use the full CEI score in the n-back analyses, but the subscores in the flanker- and cognitive task battery-analyses?

- SSA results: Although visible in both the interaction plots and results tables, I would briefly describe in words what the respective effects mean in every section where they are being presented. E.g. something like ‘only individuals who scored high on CEI show increased RTs in response to increasing payoff’; ‘only those who scored low on CEI show an increase in P3 amplitude in response to increasing payoff, whereas those with high or medium scores do not’

(11) Discussion:

- p. 34, ‘This suggests that cognitive effort investment does not negatively correlate with perceived task load (H2).’

This seems like a non-trivial finding that may deserve some more elaboration. What does this (not) indicate, e.g. does this mean the tasks are just as difficult and demanding for high scorers as for low scorers?

6. PLOS authors have the option to publish the peer review history of their article (what does this mean?). If published, this will include your full peer review and any attached files.

Reviewer #1: **Yes: **Patrick Mussel

Reviewer #2: No

---

## [Author Response · Author response to Decision Letter 0]

27 Apr 2023

Response to Reviewers

We thank the reviewers and the editor for their consideration of our manuscript and their thoughtful comments. In our revision, we have followed all advice and hope that we were able to adequately address all raised issues. Below we detail our responses (author’s responses, AR) to each specific comment made by the reviewers (RP). We believe that the manuscript has considerably been improved as a result of this process.

We highlighted all changes to the original version of the manuscript by red font in the file labeled 'Revised Manuscript with Track Changes' and provide an unmarked version labeled ‚Manuscript‘.

Reviewer #1

Thank you very much for the opportunity to review this manuscript, which I did with great pleasure. Indeed, there is a lot to like about this study: Hypotheses are grounded soundly in the literature; the design is well-planed; there are multiple tasks to generalize across task-specific effects; the sample size is relatively large; there are multiple indicators of effort investment, including subjective load, reaction time, accuracy, theta power, ERPs, and pupil dilation, and the analyses are sound. As none of my comments is "overly major", I present them in the order in which they appeared in the manuscript.

RP1.1. In the abstract and the introduction (but not later on), the authors use the concept of boredom to explain why individuals with high compared to low levels of cognitive effort investment deliberately chose harder tasks or increase their effort on demanding tasks. I doubt that the latter effect is caused by boredom avoidance. First, the absence of boredom is not necessarily cognitively demanding activity (think of simple leisure activities to kill time, such as watching television, needlework, or sitting by the fire). Second, from the early work of Berlyne to Zuckerman's sensation seeking, boredom susceptibility has been found to be a distinct dimension (e.g., from specific curiosity or experience seeking). To me, it seems more likely to consider cognitive effort investment as an approach motivation (directed towards the goal of mastering an intellectual challenge), rather than an avoidance motivation (i.e., avoiding boredom).

AR1.1. We totally agree with the reviewer in his view that

1) the absence of boredom is not necessarily cognitively demanding activity;

2) individuals with high cognitive effort investment do not differ in boredom susceptibility per se;

3) cognitive effort investment is an approach motivation rather than avoidance motivation.

We conceptualized cognitive effort investment as “self-reported dispositional differences in the willingness and tendency to exert effortful control“ [1]. Based on the observation that individuals with high cognitive effort investment tend to choose more demanding tasks, we assume that the motivation to do so comes from a perceived discrepancy between desired and actual levels of engagement, stimulation and arousal in easy tasks. Inzlicht et al. [2] proposes that this discrepancy causes the feeling of boredom. As a result, individuals with high cognitive effort investment approach a more demanding task in order to resolve this negative affect associated with boredom. 

In turn, we would not expect individuals with low cognitive effort investment to experience such a mismatch while performing easy tasks as they generally do not strive for cognitive effort. This mismatch would occur in higher demanding tasks with the opposite consequence of being overloaded with the consequence of preferring easy tasks. However, they can also be bored by easy tasks, if they are poorly rewarded. They would increase their engagement with another task, which can be as easy, in order to gather more reward [2].

As individuals with high cognitive effort investment are intrinsically motivated to engage in effortful tasks, low reward signal seems unlikely to induce a negative state associated with boredom. That is why they would not increase their engagement with a more rewarded task. 

However, we decided to remove this explanation as one of many as reviewer 2 also questioned this (RP2.1) and conceptually explained the underlying relations by the trait definitions (p. 4, ll. 56-66). We adjusted the abstract (p. 2, ll. 5-9). We hope that we were able to address this point adequately. 

RP1.2. Please give more information about how the sensitivity analysis was performed, at the very least by giving a citation - there are quite a few different ways to do so (for example, it seems that number of trials per condition was considered).

AR1.2. Given the special structure of linear mixed models, we used Optimal Design with Empirical Information (OD+) [3] for sensitivity analysis. Within this calculation program, we specified our design as cluster randomized trials with person-level outcomes with repeated measures in order to calulate power vs. effect size δ. We specified the following parameters saparately for the n-back and flanker task:

J … total sample size 

α … corrected alpha-level

n … number of trials per condition

ρ … intraclass correlation.

We now added this information right in the beginning of the procedures section (p. 10, ll. 211-213) and included it into the figure caption of S1 Fig (p. 64, ll.1195- 1196).

RP1.3. For the cognitive effort investment measures, please give some more information about the CFA (line 239). Which variables were used as indicators (items, parcels, scales, overall test scores)? 

Regarding the intellect scale, the sub-dimension of conquer seems rather an indicator of effortful self-control than cognitive motivation. 

Also, I was a bit worried that the two indicators of effortful self-control (self-control and effortful control) had no reference to epistemic behavior, but still were used as indicators of cognitive effort investment. 

The example item of the effort control scale is about sitting still while energized, which seems unrelated (or even the opposite) of the tendency to engage in cognitively challenging activities.

AR1.3. As indicators for the CFA we used the overall test scores, more specifically the normalized sum scores (as described in statistical analysis, cf. p. 20, ll. 445-448). We added this information to the description of the cognitive effort investment CFA (p.13, ll. 273- 275).

Regarding the intellect scale, we established the theoretical and empirical foundations of the core construct of cognitive effort investment in our preceding study [1]. We analyzed the correlational structure of all components (see Table 1). While conquer showed a strong correlation with need for cognition, r = .61, it has only a medium one with self-control, r = .33, and effortful control, r = .35. Therefore, we assume conquer as well as seek as valid indicators of cognitive motivation and derive it from the overall test score of intellect and do not split intellect into its subdimensions.

As to the indicators of effortful self-control, we assume a theoretical as well as an empirical association with the willingness to invest cognitive effort. Intercorrelations of cognitive motivation and effortful self-control are strong, r = .52, and even higher with cognitive effort investment, r = .87 and r = .88 [1]. Kool et al. [4] showed in their study that individuals with high self-control selected the more demanding option more often in a demand-selection task (DST) compared to a low demanding task (cf. p. 3, ll. 34-37). This illustrates the relevance of trait self-control in epistemic behavior. If we assume school and job performance as a measurable result of epistemic behavior, both self-control and need for cognition show positive correlations [5, 6] to this outcome. 

We agree with the reviewer that the chosen example item for the effortful control scale seems quite inappropriate. We exchanged this example and refer to the following one now: “If I think of something that needs to be done, I usually get right to work on it.” [“Wenn ich an etwas denke, was getan werden muss, mache ich mich gewöhnlich sofort an die Arbeit.”] (p. 12, ll. 264-266).

RP1.4. Related to the above, I would be interested to see the effects for the two subdimensions (effortful self-control and cognitive motivation). I understand that this produces a whole lot of additional tests. Describing them as additional exploratory analyses for future studies or meta-analyses (without the need for correcting for alpha inflation) or briefly summarizing any differences on the level of these dimensions, compared to the overall score (cognitive effort investment) would be a valuable additional value to the literature.

AR1.4. We recognize the additional value of these analyses for other researchers in the field and further studies and decided to include the effects for the two first-order factors in the manuscript. We added the exploratory hypotheses to the introduction (p.9, ll. 193-196), the statistical approach to the statistical analyses (p. 21, ll. 483-487), the results, that were significant and or different from that reported on the effects on cognitive effort investment, to the results section as well as a brief discussion to the first section of the discussion. We report the full results of all models in the supporting information S1 Table.

RP1.5. Regarding the NASA-TLX, did the principal component analysis support the computation of a one-factor solution (line 245)? Subjective perception of mental demands seems very close to what the authors wish to measure. on the other hand, subjective perceptions of physical and temporal demands seem much less relevant for the task at hand. Finally, subjective perception of performance and effort may even be negatively related to perceive task load (the easier the task, the better the performance). Thus, are factor scores across all six indicators the best way to assess what you aim to measure?

AR1.5. Yes. In advance to the principle component analysis, we determined the number of components for the NASA-TLX. The parallel analysis as well the scree-Test suggested 1 component. The following table shows the standardized loadings (pattern matrix) based upon correlation matrix (cf. https://osf.io/p58ub):

Scale Loading on principle component Communality Uniqueness Complexity

 PC1 h2 u2 com

effort 0.82 0.68 0.32 1

frustation 0.82 0.67 0.33 1

time 0.82 0.67 0.33 1

mental 0.77 0.59 0.41 1

performance 0.65 0.43 0.57 1

physical 0.43 0.18 0.82 1

All scales show good loading on the general factor „NASA-TLX“. For a closer look, the communality column shows the variance of the scales that is explained by the general factor „NASA-TLX“, whereas the uniqueness column indicates the proportion of variance that is not explained by the general factor. In total, the variance of all scales seems to be well explained by the general factor. However, as anticipated by the reviewer, physical load shows a worse fit and, therefore, will be less represented by the one factor solution. From our point of view, this is conceptually acceptable as we do not expect extraordinary physical load during the performance of the computer tasks and as physical load is not the main focus of our study. Considering the alternatives, e.g. determining simple mean or median, aggregating a selection of scales, we deem PCA the best way of aggregation.

We added our considerations to the manuscript (p. 13, ll. 282-285):

„The theoretically assumed one-factor solution for perceived task load was confirmed by both parallel-analysis and scree-Test. All scales loaded highly on this factor; standardized loadings ranged from 0.43 for physical to 0.82 for effort, frustration and time.“.

RP1.6. Regarding the description of the n-back task (line 289), the authors first note four levels of demands (0-back, 1-back, 2-back, 3-back), but later on only three levels (level (0, 1, 2), line 304).

AR1.6. Thank you very much for pointing this out – that is a mistake. It should be “0-back, 1-back, 2-back“ instead.

The decision for three n-back levels is line with Mussel et al. [7]. In our pilot studies, higher n-back levels (as used in [8]) turned out to evoke a high frustration and low commitment in participants.

RP1.7. Regarding the Flanker tasks, the author mention six blocks, but only three seem necessary for manipulating payoff (0.01, 0.02, 0.04 EUR). Why are there six blocks?

AR1.7. The design of the flanker task based on the one used in a study of Forsteret al. [9] who investigated the N2-ERP-component as conflict signal in a modified flanker task. Within this task, levels of response conflict vary parametrically, i.e. demand levels increase with the number of incongruent distractor stimuli. In its original setup, the paradigm consisted of six blocks in the absence of reward. In our study, we kept the six blocks and distributed the three different payoffs to two blocks each. We randomized the order of payoff blocks to avoid sequence effects and hence, to account for fatigue. To clarify the block design, we changed to order of the descriptions for both tasks (cf. RP2.19) and specified the assignment of payoff levels to blocks (p.15, ll. 355-357). 

RP1.8. Any reason why the Gratton & Coles procedure to remove blink artefacts is used, rather than more recent approaches like ICA (see 10.3389/fnins.2021.660449).

AR1.8. We are aware of other ocular correction methods like ICA and we, in general, agree that ICA might be the method of choice. However, we are relying on the options that come with BrainVision Analyzer. We made the experience that ICA solutions in BrainVision Analyzer still has some weaknesses (and made data quality even sometimes worse), whereas the Gratton & Coles procedure works very well in removing blink artifacts. Thus, we had agreed on the Gratton & Coles procedure in our lab. We added our motivation on using this procedure to the manuscript (p. 17, l. 382).

RP1.9. Please specify how the windows for the ERPs were specified (peak detection on the grand mean?). Why was such a small window chosen for the large P3-component?

AR1.9. We specified the N2 time window a priori based on the trade-off between specifity and sensitivity (i.e., narrower vs. longer time window) and followed Forster et al. [9], who defined the N2 peak around 310 ms after target presentation. We have waived an adjustment of this peak based on individual’s average latency and assumed a time window of 310 ± 20 ms from target onset appropriate. Our theoretical considerations proved to fit very well with our data facing specifity and sensitivity (cf. Fig 7). As to the P3, its peak is generally expected between 250 - 600 ms poststimulus. We chose 400 ms as the midpoint and same length of ± 20 ms to keep the definitions similar. This time window has also been confirmed in the data (cf. Fig 8). We added a comment on that in the manuscript (p. 18, ll. 394-395).

RP1.10. Line 439: The authors note that they centered all level 1 predictors within person. If I understood correctly then level 1 predictors are "demand" and "payoff", i.e., experimentally controlled variables which should have equal means across participants anyway. Therefore, why center within persons?

AR1.10. Yes that is correct, demand and payoff are level 1 predictors (besides correct and postcorrect responses, late responses (flanker task) and block). Although both are experimentally controlled variables, they do not have equal means across participants. As to task design (i.e., distribution of trials within conditions), there are 196 trials per payoff level and 411 trials in the low demand condition as well as 59 trials in each other demand condition in the flanker task. Indeed, we have 72 trials in each parcel (payoff × demand) of the n-back task. However, the number of trials varies for each individual due to missing/ excluded values during recording and preprocessing (cf. p. 20, ll. 448-450). 

RP1.11. This might be a matter of debate, but I wondered whether corrections for multiple testing were overly strict. If I understood correctly, results are corrected family wise for seven independent tests according to the seven indicators of cognitive effort (i.e., reaction time, accuracy, early and late theta power, N2 and P3, and pupil dilation). Corrections for multiple testing are appropriate whenever multiple tests are performed to test one substantial hypothesis. As the hypotheses are formulated for "effort indices", the authors' approach is generally appropriate. It implies that a hypothesis is confirmed whenever one of these tests is significant on the adjusted significance level. However, from a more substantively point of view, this sems very strict. The approach corresponds with the assumption of response coherence, i.e., effects of personality traits are equivalent across responses (behavioral, psychophysiological, etc.). It is well known that this is not the case as effects are strongly depending on the respective response (e.g., Lang, 1971; Asendorpf, 1988). Thus, one could also refrain from corrections for multiple testing, while interpreting a significant effect as confirming the hypothesis for this particular response, rather than cognitive effort in general (i.e., across indices).

AR1.11. We agree that it is a matter of debate and that our assumption of response coherence is very idealistic. Finally, we find your remarks very convincing and omitted corrections for multiple testing. We made a comment on the absence of corrections (p.20, ll. 450-452) and make clearer that the presented hypothesis refer to each index instead across indices (p. 9, ll. 180-191). Moreover, we adjusted the tables, the results and headings in the results section where necessary. 

RP1.12. According to the (selective) results presented for levels of demand on responses in the Flanker task (see Figures 7 and 8), I wondered whether it is appropriate to use "demand" as a continuous predictor. Levels of demands (especially "0%") seem to have qualitatively different effects. This suggests that a categorical variable might be more appropriate.

AR1.12. Indeed, this might be a matter of debate. Linearity might vary with demand implementation in different tasks and even with different effort indices. We carefully thought that through and generally assume linearity as this seem to be the case in the underlying studies [7-10]. We have decided against demand as factor, as analysis and interpretation would be more complicated. 

Regarding Figures 7 and 8, interpretation of raw EEG values is to be taken with caution. Both figures display raw means averaged across all participants and all blocks. During statistical analysis this raw data is statistically cleaned (e.g., nesting of data within the multi-level model, separation of main and interaction effects) as well as controlled for possible influences (e.g., correct/ postcorrect responses, block number, etc.). Thus, these figures can and should only be used to visually evaluate the quality of the EEG signal, but not to assess possible effects or relations between conditions. 

We now point that out when we introduce these figures (p. 36, ll. 632, 637-638 and p. 38, ll. 656, 663-664). 

RP1.13. The discussion is mainly about why the expected effects were not found. Maybe related to the above issue of corrections for multiple testing, I found this perspective to be one-sided and very strict. The interaction effects between CEI and payoff for reaction times are well in line with Deci and Ryan's self-determination theory which predicts positive effects of incentives to have positive effects for externally motivated individuals (which corresponds to low levels of CEI) and no or even detrimental effects for intrinsically motivated individuals (corresponding to high levels of CEI) as they undermine intrinsic motivation. Results for P3-amplitudes (Fig. 9) show a neural correlate of this effect. Additionally, the significant interaction between CEI and demands in the Flanker task on early theta is well in line with predictions by Cacioppo who predicted stronger increases in cognitive effort for high compared to low NFC individuals as demands increase, and mirrors results from Mussel et al. for late theta. Also, there was a significant effect of CEI on accuracy in the n-back task that was not further interpreted as not being robust when controlling for multiple testing. In my opinion, these results merit to be acknowledged.

AR1.13. In the course of omitting the correction for multiple testing, we worked on the discussion of the results and made changes to the manuscript (see discussion section). 

RP1.14. When discussing differential effects for n-back and Flanker task (e.g. page 39), you may also mention block-wise vs. trial-wise manipulation of demands in the two tasks which may have profound effects on the results (e.g., https://doi.org/10.1093/scan/nsab126).

AR1.14. We thank the reviewer for this additional consideration of further explanations regarding the differential effects for n-back and flanker task. We now mention it in the limitations section (p. 47, ll. 816-818). 

RP1.15. Check whether you can avoid some abbreviations that are rarely used to reduce the cognitive load for (especially low CEI-) readers, e.g., SSA, ICC, FDR.

AR1.15. We thank the reviewer for this hint and are happy to reduce the cognitive load for all readers by reducing these abbreviations (i.e., SSA, ICC, FDR, CEI, COM, ESC).

Reviewer #2

**** SUMMARY & GENERAL COMMENTS: ****

This study investigates to what extent dispositional scores on a newly composed cognitive effort investment scale are related to a range of behavioral and psychophysiological markers of mental effort and whether they moderate the effect of task demand and payoff on mental effort exertion.

This is a thorough and interesting study with well documented methods, including the openly available data and analyses scripts. My main concerns regard clarification (and justification) of the specific predictions and operationalization of mental effort with the different measures used in this study. Furthermore, while I appreciate the multidimensional approach, it makes it somewhat difficult to synthesize the findings. I feel that the authors’ motivation to do this, their predictions regarding the specific measures, and their expectations regarding the overall picture of findings (across measures) could be specified in a bit more detail in the introduction.

Detailed comments on this and other issues are outlined below.

**** MAJOR COMMENTS: ****

(1) The predictions regarding moderation of demand and payoff effects on effort by interindividual differences in the cognitive effort investment trait component (CEI) need more explanation and justification (particularly p.3-4 & Fig. 2).

(1a) Regarding demand:

RP2.1. If individuals scoring high on CEI increase their actual effort investment more with increasing task demands than those scoring low, this may be due to a number of reasons. They may have more cognitive resources available, and/or be better at detecting demand-differences and allocate effort accordingly, and/or be more motivated to try and thus not give up as easily as those scoring low, etc. Maybe the authors could specify what they assume might underlie such a moderation effect; especially given their CEI component includes aspects of both motivation and self-control.

AR2.1. Indeed, there are many possible explanations why individuals with high cognitive effort investment increase their effort stronger with task demands. Finally, it has been observed that individuals with high need for cognition (being a main component of cognitive effort investment) dynamically allocate cognitive resources indicated by frontal midline theta power, whereas individuals with low need for cognition invested a considerably higher amount even in easier tasks [7]. This empirical evidence supports the theoretical conceptualization of not only the indicator variables but also cognitive effort investment itself. Individuals scoring high on these dimensions, e.g., approach and enjoy cognitive endeavors [11] and perform self-controlled behavior relatively effortlessly and unconsciously [6]. However, there is no evidence for increased cognitive resources in individuals with high need for cognition but for higher motivation to perform effortful tasks [12]. Hence, these individuals associate a higher the subjective value with demanding tasks than individuals with low need for cognition do [8]. 

Taken together with RP1.1., we decided to re-write this part in the introduction and hope to be more precise on this point, now (p. 4, ll. 56-74).

Although, we agree that demand sensitivity might be an interesting alternative concept to be investigated in further studies, we had no a priori hypothesis in this regard.

RP2.2. The general relationship between task demands and actual effort investment may not be linear (as presented here, Fig.2) but follow an inverted U-shape, where, as demands exceed resources, detachment from the task at hand occurs and effort decreases (see e.g. van Steenbergen et al., 2015, doi: https://doi.org/10.3389/fpsyg.2015.00974). In theory, it could then be expected that interindividual differences are not necessarily in slope, but in ‘turning point’ (peak of the inverted U-shape). To what extent such a non-linear pattern emerges may depend on the task at hand and the different demand levels included, and might thus not be expected for the tasks used in this study. Nevertheless, the authors may want to consider this and mention it briefly when presenting their linear theoretical model.

AR2.2. The linear relationship between task demands and actual effort investment can indeed be questioned (cf. RP1.12). We carefully thought that through and generally assume linearity as this seem to be the case in the underpinning studies [7-10].

It seems that the referenced paper likewise supports a linear relationship more than an inverted u-shape one. Van Steenbergen et al. 2015 investigated the relationship between perceived task difficulty (compared to demand manipulation) and conflict adaptation in the flanker (two demand levels) and Stroop task (2 demand levels) within four experiments. In experiment 1 (N = 91) and 2 (N = 28) they found no reliable association, a negative linear relation in experiment 3 (N = 27) in the flanker task only and a positive linear relation in the flanker task in the fourth experiment (N = 198). To sum up, their evidence in favor of an inverted U-shape relationship is rather not supported. 

In theory, one could imagine an inverted U-shape if task demands exceed abilities. Indeed, we observed high frustration and low commitment in participants during n-back levels n > 2 in our pilot study. Therefore, we implemented n-back levels 0,1,2 instead of 1 – 6 back like in the study by Westbrook et al. [8]. 

Regardless of these considerations, we are happy to discuss the linearity in the discussion (p. 47, ll. 799-800, 830-835).

 (1.b) Regarding payoff:

RP2.3. Particularly the prediction that high CEI scorers would increase actual effort less with increasing payoff needs more justification. Why would they care less about higher payoffs than lower scoring individuals? Again, multiple interpretations are possible, and I hope the authors can clarify their standpoint on this. 

• It may be that high scorers don’t have to increase their effort with payoff because they are ‘already giving it all’ at low payoffs. This may also be demand dependent: at high demands, it might be more likely for low scorers not to increase their effort investment with increasing payoffs, as it may not be worth it (‘giving up’), whereas high scorers may still be ‘up for the challenge’ - this would result in opposite effects than those predicted here. 

• Another possibility, and perhaps closest to the authors’ interpretation, is that low and high scorers differ in their motivation to participate in the study to begin with. While low scorers may be motivated by the potential rewards, high scorers may participate mostly because they enjoy such tasks.

AR2.3. We argue in line with the second interpretation. Justification for this derived on the one hand from theoretical conceptualization of the underlying personality traits and on the other hand on empirical findings of Sandra and Otto [13].

As to cognitive effort investment, individuals with high scores are highly motivated to engage in cognitively demanding tasks [11] independently from external motivational incentives, whereas individuals with low scores tend to avoid cognitive effort [4] and get motivated by external reward. 

These assumptions are supported by findings of Sandra and Otto [13] showing a negative relation between need for cognition and reward-induced performance changes. That means, individuals with low need for cognition showed a stronger performance improvement by higher reward, whereas individuals with high need for cognition showed no improvement or even a decrease. 

We re-wrote the respective part in the introduction and hope to be more precise on this point, now (cf. RP2.1; p. 4, ll. 56-74).

RP2.4. Note here also a lack of clarity regarding the following formulation (p.4): ‘They do not benefit from payoff in terms of improved performance or recruiting more resources in order to improve performance.’ – What do the authors mean by ‘benefit’?

AR2.4. This statement refers to the preceding two sentences and just sums up what have been said. The word ‘benefit’ is used as a general expression that fits all changes in the different effort indices, as they do not all increase or decrease, but have in common that they refer to an advantage in task processing. We re-formulated the sentence and added two examples and hope to be more precise now (p. 4, ll. 69-73). 

(1.a+b) The supporting evidence cited for both predictions (p.4, l.64-70) should also be explained with a bit more detail:

RP2.5. The Mussel, Ulrich [12] study found that individuals with high levels of need for cognition allocate their cognitive resources according to task demands. But what about those with low levels? Only if they differ would this support the above-mentioned prediction.

AR2.5. We apologize for not being as precise in this regard and added the findings on individuals with low levels of need for cognition (p. 5, ll. 80-82).

RP2.6. The Sandra and Otto paper [13] should also be discussed more carefully, as it (unexpectedly) shows not a lower adaptation of effort to payoff in those with high need for cognition, but a negative one. This is not in line with predictions by the authors of the current study.

AR2.6. Sandra and Otto [13] present a negative relation between need for cognition and reward-induced performance changes (significant reward x need for cognition interaction). That means, individuals with low need for cognition showed a stronger performance improvement by higher reward, whereas individuals with high need for cognition showed no improvement. 

Although the authors plotted performance (i.e., task-switch costs in RT) for low and high need for cognition as a function of reward condition, they did not report on the significance of the descriptive differences. One could now assume a significant decrease of switch costs by high reward for low need for cognition scores. However, the picture is more ambiguous for high need for cognition scores. Consequently, we deleted the statement of a possible decrease, as there is not enough evidence (p. 5, l. 84). 

(2) The role of cognitive ability:

RP2.7. In several sections of the paper, it is stated that individuals with high levels of dispositional CEI are ‘under-stimulated by easy tasks’, ‘find task demands easier’, ‘exert effort more efficiently’, etc. This suggests a potential link to cognitive ability/resources, which is only briefly (and comparatively late; p.7, l.141-151) mentioned in the introduction where it stands somewhat in isolation. In contrast, the statements mentioned above seem to be scattered somewhat unsystematically throughout the paper, not directly linked to the question of cognitive ability/resources, and it is a bit unclear what the authors meant by this and (in part) how they expect this would translate into findings of their study (especially regarding ‘exert effort more efficiently’) .

I think the question of whether effort investment differences may be related to differences in cognitive ability/resources should be addressed early and more systematically in the introduction, and it should be made clear whether statements such as ‘under-stimulation’ or ‘efficient effort allocation’ are meant to be related to this.

AR2.7. We thank the reviewer for the suggestions and moved a revised paragraph on cognitive abilities more to the beginning of the introduction to p. 5, ll. 85-101. 

(3) Accuracy, reaction times, and allocating effort ‘efficiently’:

RP2.8. There is a difference between investing effort per se and investing effort efficiently. This paper seems to refer to one concept in some and the other in other sections. It needs to be made clearer what the focus is, whether interindividual differences are expected for both or just one of the constructs, and how this would be reflected in the different effort indices.

AR2.8. We re-wrote the parts of the introduction and hope to be more precise on this, now (cf. RP2.1, RP2.3; e.g. p. 4, ll. 56-74).

RP2.9. When introducing the different measures of effort, most of the evidence provided to justify them as ‘effort measures’ regards their association with task load/demand. Naturally, effort should be higher under higher demands, i.e. it is reasonable to assume that these measures also reflect invested effort. However, especially for the behavioral measures, that is not necessarily the case. If accuracy is low, one cannot know whether not enough effort was exerted, or whether all the effort was exerted, but not successfully. Regarding reaction times (RTs), the presented picture is even more unclear. They are first presented as ‘longer under high task load’, which suggests that more effort should translate into longer RTs. But in their own study, the authors take shorter reaction times to reflect ‘successfully invested effort’, and longer to reflect lower effort (similar to the cited Sandra and Otto study [13]).

AR2.9. In general, we agree that accuracy decreases, if task demands exceed abilities. We are confident that this was not the case in the discussed studies as task demand were comparably moderate (e.g. n-back task with levels 0 to 2 [7]). 

As to reaction time, the harder the task, the longer the reaction time. That this relation is diminished when presented with monetary incentives supports the assumption that with higher effort investment, reaction time decreases. In other words, if one invests the same effort to tasks with different demands, reaction time gets slower with increasing demand; and if one invests more effort to task processing, reaction times get faster. Thus longer reaction time translates to “less invested effort”.

(4) Results:

RP2.10. Make clear which (and how) analyses were corrected for cognitive abilities and crystallized intelligence and report the associated findings (e.g. main effects).

AR2.10. As stated in the statistical analyses section (cf. pp.21), we controlled all mixed effects models for cognitive abilities and crystallized intelligence by including them into the mixed effects models as additional predictors, i.e., continuous level 2 predictor. We added a brief note on that (p. 21, l. 464).

Within the paper we refrained from reporting all control variables due to reasons of space and the fact that we statistically eliminated any possible influence on the presented results. We agree with the author on the importance of reporting the associated findings on cognitive abilities and crystallized intelligence and decided to include a separate, brief paragraph in the results section that presents the main effects of cognitive abilities and crystallized intelligence (p. 23, ll. 507-515).

The more interested reader is invited to have a look into our analysis scripts openly stored at OSF. The commented-out script not only contains all analysis but additionally the main results. 

(5) Methods:

RP2.11. p.17, linear mixed effects model: Why does the model include a 3-way interaction when this is not a question the authors set out to answer? I do believe this interaction is relevant to fully understand the interplay between demand- and payoff-motivated effort investment, and the moderation thereof by interindividual differences/preferences. However, given this question was not one the authors addressed in their introduction, its inclusion here is a bit surprising.

I would suggest either to update introduction and predictions accordingly, or to restrict the model to those 2-way interactions actually relevant to the hypotheses.

AR2.11. We totally agree with the reviewer that it would be statistically sounder if we would exclude the three-way-interactions for the models, as we do not test any hypothesis on this regard. We considered this option in advance and discussed it again. 

Apart from the fact that a reliable evidence in the existing literature is missing as a basis for hypothesizing on these three-way-interactions, they are additionally hardly interpreted. Nevertheless, we believe in the added value for the scientific community by reporting these interactions (similar to reporting the results of the subdimensions of cognitive effort investment as exploratory analysis as proposed by reviewer 1, cf. RP1.4). We now highlight the exploratory nature of the three-way interactions (p. 9, ll. 192-193) 

(6) Discussion:

RP2.12. p.35, l.659-662: It is unclear that (and why) the here presented RT results were predicted (see also general comment above).

The RT-findings should be interpreted carefully and discussed more thoroughly.

Note that higher demands lead to longer RTs in both tasks, and higher payoff to longer RTs in the nback- but shorter in the flanker-task. This suggests a different association of RTs with effort in the different tasks. Given the flanker-task is more fast paced (response time limit = 1200ms), maybe the speed-accuracy trade-off is more prominent here? I.e., one may ‘risk’ getting it wrong after shorter deliberation and focus on ensuring one responds in time when pay-offs are high.

AR2.12. Please consider AR2.9., as this point also refers to the same. Fig 5 in the manuscript displays the interaction of reaction time and payoff while indicating different slopes for high, medium and low levels in cognitive effort investment. The slope for high levels shows in increase according to payoff in both task, while the slope in the n-back task is much more steeper. The pattern for individuals with low levels is the opposite, their reaction times get faster with increasing payoff. Both results are in line with our predictions and explained in detail in the discussion (cf. p. 46, ll. 754-763).

RP2.13. p.40, ‘Taken together, our results do not support the notion that individuals with high levels of cognitive effort investment exert effort more efficiently.’

It is unclear that and why this was expected and how this ‘efficiency’ was operationalized with the different effort indices (see general comment above).

AR2.13. We re-formulated the respective part in the introduction and hope to make that clearer now p. 4, ll. 56-74).

**** MINOR COMMENTS: ****

(7) 

RP2.14. The term ‘effort discounting’ is used a bit imprecisely throughout the manuscript (e.g. p.3, l.28-29). Effort discounting typically describes the principle that the subjective value of a reward decreases as the amount of effort required to obtain it increases (e.g. p.16 in [2], i.e. Botvinick et al., 2009). Instead, the authors here seem to use the term interchangeably with the ‘law of least mental effort’.

AR2.14. We were not aware that this happened and are sorry for the confusion. However, we checked the use of the term “effort discounting” and identified the one that could have been misleading and specified accordingly (p. 3, ll. 49-50 and p. 44, l. 713).

(8) Introduction:

RP2.15. p.5, l.100-102: Explain that switch costs here reflect RTs and that here (unlike in the second sentence of the same paragraph) one assumes shorter RTs reflect more (efficiently allocated) effort (see also major comments on RTs).

AR2.15. We added a sentence that explains the interpretation of switch costs (p.7, ll. 129-132).

RP2.17. p.7, (H2): This hypothesis has not been well prepared/justified in the introduction. The authors should add a short explanation on why they assume perceived task load is lower in high CEI scorers (see also major comments on CEI-based predictions and role of cognitive abilities).

AR2.16. We re-wrote the parts of the introduction and hope to be more precise on this, now (cf. RP2.1, RP2.3; e.g. p. 4, ll. 56-74).

(9) Methods:

RP2.18. ‘Participants completed 20 training trials for each difficulty level until they completed at least 85 % trials correctly.’:

Does that mean 20 training trials per difficulty level were enough for all participants to reach the 85% correct criterion, or were sets of 20 trials per difficulty level repeated until this criterion was reached? If the latter: consider providing total number of required training trials per task and testing whether this was associated with CEI.

AR2.18. Participants had the possibility to repeat the set of 20 training trials in both tasks in order to reach the 85 % correct criterion. We hope that we are now clearer on this (p. 15, l. 330 and p. 16, l. 354).

The reviewer’s questions on the total number of training sets is an interesting one and would be nice to investigate. However, we had no a priori hypotheses regarding this question. Thus, it is a welcome question for further studies. 

RP2.19. It seems blocks differed by demand in the n-back but by payoff in the Flanker task? This block design could be explained a bit more clearly in the descriptions of the tasks, and prior to stating things like ‘at the end of each block’.

AR2.19. We changed the order of description for both tasks and hope to be clearer on the block design now (p. 15, ll. 330-336 and p. 16, ll. 354-360).

RP2.20. Pupil size preprocessing: if I understood correctly, all eye-blinks and other artifacts were treated as missing, and not interpolated. This may lead to a lot of data loss, but it also makes the remaining pupil size measures somewhat more reliable (reflecting the ‘real’ pupil size), so I don’t argue against this approach per se. However, I wonder how much data was lost overall. When the authors state that ‘Trials with missing data, set to NA or with RTs < 100 ms were excluded from statistical analysis’ (p. 17), does that mean that pupil size averages for the time windows of interest were not calculated if any samples within the respective time window were missing? It would be good to report how many trials after data exclusion were available per participant and per task in the end.

AR2.20. We added a statement on the percentage of missing trials as well as mean and range on average per participant in the end of this paragraph (pp. 19, ll 438-440). 

No, it does not mean, “that pupil size averages for the time windows of interest were not calculated if any samples within the respective time window were missing”, but that we have excluded them from the calculation of pupil size in the respective time window. As we treated missing data as NA, they have no impact on the mean calculation. However, here we state, that trials resulting in NA were not included in statistical analyses, i.e. correlation analyses and LMMs and GLMMs. 

RP2.21. p.17, l. 406-407: I believe the reference here should be [56], not [29]

AR2.20. We are grateful for the reviewer’s attentive reading. Obviously we have mixed up the references as they share the first author’s name as well as year of publication. We resolved this mistake by adding the right reference on p. 20, l. 436. 

(10) Results:

RP2.22. Use decimal points (not comma) in all presented numbers (see tables)

AR2.22. We reviewed all tables and made the changes where appropriate (i.e., Tables 6, 7, 9). 

RP2.23. p.19, ‘Manipulation check: Effort indices reflect levels of demand and partly of payoff’:

Maybe state explicitly that the p-values presented here refer to the main effects presented in the tables 2-9. I would also add a short note on the direction of the effects.

AR2.23. We added a reference to the respective tables (p. 22, ll. 504, 506). However, we do not indicate the directions of the effects due to space restrictions as this information is not decisive to our hypotheses and as it is easily accessible within the tables). 

RP2.24. p.20, ‘Perceived task load: No relation to cognitive effort investment’:

Why use the full CEI score in the n-back analyses, but the subscores in the flanker- and cognitive task battery-analyses?

AR2.24. We thank the reviewer for this point. We were not aware of the inconsistency. However, with regard to RP1.4, we decided to report the effects for the subdimensions (i.e., cognitive motivation and effortful self-control) of cognitive effort investment as additional exploratory analyses for future studies or meta-analyses. We included the exploratory nature within our hypothesis (p. 9, ll. 192-196). 

RP2.25. SSA results: Although visible in both the interaction plots and results tables, I would briefly describe in words what the respective effects mean in every section where they are being presented. E.g. something like ‘only individuals who scored high on CEI show increased RTs in response to increasing payoff’; ‘only those who scored low on CEI show an increase in P3 amplitude in response to increasing payoff, whereas those with high or medium scores do not’

AR2.25. We are happy for this point and added a brief explanation of the simple slope results to the sections where they are presented, i.e. reaction time (p. 28, ll. 551-554, 566-568), early FMθ power (p. 34, ll. 608-610), and P3 amplitude (p. 41, ll. 676-678).

(11) Discussion:

RP2.26. p. 34, ‘This suggests that cognitive effort investment does not negatively correlate with perceived task load (H2).’

This seems like a non-trivial finding that may deserve some more elaboration. What does this (not) indicate, e.g. does this mean the tasks are just as difficult and demanding for high scorers as for low scorers?

AR2.26. This indicates that individuals perceived task load equally and independently of their level in cognitive effort investment. We added this explanation to the manuscript (p. 45, ll. 737-738).

References

1. Kührt C, Pannasch S, Kiebel SJ, Strobel A. Dispositional individual differences in cognitive effort investment: establishing the core construct. BMC Psychol. 2021;9(1):10. Epub 2021/01/24. doi: 10.1186/s40359-021-00512-x. PubMed PMID: 33482925; PubMed Central PMCID: PMCPMC7821547.

2. Inzlicht M, Shenhav A, Olivola CY. The Effort Paradox: Effort Is Both Costly and Valued. Trends Cogn Sci. 2018;22(4):337-49. Epub 2018/02/27. doi: 10.1016/j.tics.2018.01.007. PubMed PMID: 29477776; PubMed Central PMCID: PMCPMC6172040.

3. Raudenbush SW, Spybrook J, Congdon R, Liu X, Martinez A, Bloom H, et al. Optimal Design Plus Empirical Evidence. 3.0 ed2011.

4. Kool W, McGuire JT, Wang GJ, Botvinick MM. Neural and behavioral evidence for an intrinsic cost of self-control. PLoS One. 2013;8(8):e72626. Epub 2013/09/10. doi: 10.1371/journal.pone.0072626. PubMed PMID: 24013455; PubMed Central PMCID: PMCPMC3754929.

5. Bertrams A, Dickhäuser O. High-school students' need for cognition, self-control capacity, and school achievement: Testing a mediation hypothesis. Learn Individ Differ. 2009;19(1):135-8. doi: 10.1016/j.lindif.2008.06.005.

6. de Ridder DTD, Lensvelt-Mulders G, Finkenauer C, Stok FM, Baumeister RF. Taking Stock of Self-Control. Pers Soc Psychol Rev. 2011;16(1):76-99. doi: 10.1177/1088868311418749.

7. Mussel P, Ulrich N, Allen JJ, Osinsky R, Hewig J. Patterns of theta oscillation reflect the neural basis of individual differences in epistemic motivation. Sci Rep. 2016;6:29245. Epub 2016/07/07. doi: 10.1038/srep29245. PubMed PMID: 27380648; PubMed Central PMCID: PMCPMC4933953.

8. Westbrook A, Kester D, Braver TS. What is the subjective cost of cognitive effort? Load, trait, and aging effects revealed by economic preference. PLoS One. 2013;8(7):e68210. Epub 2013/07/31. doi: 10.1371/journal.pone.0068210. PubMed PMID: 23894295; PubMed Central PMCID: PMCPMC3718823.

9. Forster SE, Carter CS, Cohen JD, Cho RY. Parametric manipulation of the conflict signal and control-state adaptation. J Cogn Neurosci. 2011;23(4):923-35. Epub 2010/02/12. doi: 10.1162/jocn.2010.21458. PubMed PMID: 20146615; PubMed Central PMCID: PMCPMC3056266.

10. Scharinger C, Soutschek A, Schubert T, Gerjets P. When flanker meets the n-back: What EEG and pupil dilation data reveal about the interplay between the two central-executive working memory functions inhibition and updating. Psychophysiology. 2015;52(10):1293-304. Epub 2015/08/05. doi: 10.1111/psyp.12500. PubMed PMID: 26238380.

11. Cacioppo JT, Petty, R. E.; Feinstein, J. A.; Jarvis, W. B. G. Dispositional Differences in Cogntive Motivation: The Life and Times of Individuals Varying in Need for Cognition. Psychol Bull. 1996;119(2):197-253.

12. Gärtner A, Grass J, Wolff M, Goschke T, Strobel A, Strobel A. No relation of Need for Cognition to basic executive functions. J Pers. 2021;89(6):1113-25. Epub 2021/04/19. doi: 10.1111/jopy.12639. PubMed PMID: 33866562.

13. Sandra DA, Otto AR. Cognitive capacity limitations and Need for Cognition differentially predict reward-induced cognitive effort expenditure. Cognition. 2018;172:101-6. Epub 2017/12/17. doi: 10.1016/j.cognition.2017.12.004. PubMed PMID: 29247878.

---

## [Decision Letter · Decision Letter 1]

30 May 2023

PONE-D-23-01823R1Cognitive effort investment: Does disposition become action?PLOS ONE

Dear Dr. Kührt, Thank you for submitting your manuscript to PLOS ONE. After careful consideration, we feel that it has merit but does not fully meet PLOS ONE’s publication criteria as it currently stands. Therefore, we invite you to submit a revised version of the manuscript that addresses the points raised during the review process.

We look forward to receiving your revised manuscript.

Kind regards,

Árpád Csathó, Ph.D.

Academic Editor

PLOS ONE

Journal Requirements:

**Additional Editor Comments:**

Both reviewers have submitted their review about your revised manuscript. Both gave clearly positive feedback; and I also think that the manuscript has been significantly improved. However, one reviewer still suggested some minor changes. Before accepting your manuscript, I would like to invite you to address these minor suggestions raised by the Reviewer in a next revision.

Reviewers' comments:

Reviewer's Responses to Questions

**Comments to the Author**

1. If the authors have adequately addressed your comments raised in a previous round of review and you feel that this manuscript is now acceptable for publication, you may indicate that here to bypass the “Comments to the Author” section, enter your conflict of interest statement in the “Confidential to Editor” section, and submit your "Accept" recommendation.

Reviewer #1: All comments have been addressed

Reviewer #2: (No Response)

2. Is the manuscript technically sound, and do the data support the conclusions?

Reviewer #1: Yes

Reviewer #2: Yes

3. Has the statistical analysis been performed appropriately and rigorously? 

Reviewer #1: Yes

Reviewer #2: Yes

4. Have the authors made all data underlying the findings in their manuscript fully available?

Reviewer #1: Yes

Reviewer #2: Yes

5. Is the manuscript presented in an intelligible fashion and written in standard English?

Reviewer #1: Yes

Reviewer #2: Yes

6. Review Comments to the Author

Reviewer #1: The authors were very responsive and have addressed all issues raised during the revision process. I congratulate them on a fine manuscript!

Reviewer #2: I think the manuscript has been improved greatly. Some minor points could still profit from clarification, just to improve consistency and logical flow of an else really well written paper.

Page and line numbers below refer to the version with highlighted changes, and text snippets from the manuscript are included for easier review.

###### Introduction: ######

(1) p.3, l.33-34: ‘The general tendency of avoiding or at least reducing cognitive work – or effort – is called effort discounting [2].’

Comment: This definition of effort discounting is still imprecise (as it is used interchangeably with the ‘law of least mental effort’) and not exactly in line with the reference [2] provided here; see first round of review. It might be best to simply remove this sentence.

(2) p.4, l.68-74: ‘(…) Westbrook et al. [6] experimentally showed that individuals with high need for cognition (i.e., a main indicator of cognitive effort investment) indicated a higher subjective value with demanding tasks than individuals with low need for cognition. Therefore, individuals with high levels in cognitive effort investment would be less likely to increase their effort based on increasing payoff, but rather based on increasing demand. In other words, when given equal demands, individuals with high compared to low levels of cognitive effort investment exert less effort.’

Comment: I appreciate the added section which this paragraph is a part of, with a good explanation of what is meant by efficient effort allocation and evidence presented for the intrinsic motivation of individuals with a high CEI disposition to perform effortful tasks. However, the conclusions in the last part do not strictly follow from the evidence presented. The evidence presented shows that those with high NFC seem to like demanding tasks more than those with low NFC. But this does neither (‘therefore’) indicate that the high-scorers would be less likely to increase their effort based on increasing payoff, nor (‘in other words’) that at equal demands high scorers would exert less effort. At least not directly – more evidence is needed to make this case and these predictions.

(3) p.6, l.134-136: ‘In line with our assumption that individuals with high levels of cognitive effort investment find task demands easier compared to individuals with low levels (…)’

Comment: At this point in the paper, this assumption appears a bit surprising as it has not been properly introduced and substantiated with evidence yet. While the authors earlier acknowledge the correlation between e.g. NFC and different cognitive ability/capacity measures, they have thus far focused on justifying why associations between NFC and demand- and payoff-related effort-adjustment are not due to differences in cognitive ability. But they have not clearly stated that they assume task demands will be perceived as easier by those scoring high on CEI.

###### Discussion: ######

(4) p. 44, l. 786-790: ‘In the present study, we were interested in the processes underlying this motivation to seek out effort. We focused on the main questions: (1) Is cognitive effort investment related to objective effort indices? (2) Does cognitive effort investment moderate the relation between effort indices and demand and payoff?’

Comment: Strictly speaking, there is no (direct) relation between the presented hypotheses and the interest in ‘the processes underlying this motivation to seek out effort’. Seeing the other revisions that were made, I am assuming the first sentence here was supposed to be changed as well.

(5) p.45, l.801-804: ‘Hence, individuals with high levels of effortful self-control generally experience lower effort during task processing, which is consistent with the findings of de Ridder et al. [12] and strengthens our assumptions on individuals scoring high on cognitive effort investment.’

Comment: The last sentence sounds somewhat incomplete – which assumptions exactly does this strengthen?

7. PLOS authors have the option to publish the peer review history of their article (what does this mean?). If published, this will include your full peer review and any attached files.

Reviewer #1: **Yes: **Patrick Mussel

Reviewer #2: No

---

## [Author Response · Author response to Decision Letter 1]

29 Jun 2023

Reviewer #1

The authors were very responsive and have addressed all issues raised during the revision process. I congratulate them on a fine manuscript!

Reviewer #2

I think the manuscript has been improved greatly. Some minor points could still profit from clarification, just to improve consistency and logical flow of an else really well written paper.

Page and line numbers below refer to the version with highlighted changes, and text snippets from the manuscript are included for easier review.

###### Introduction: ######

RP2.1. p.3, l.33-34: ‘The general tendency of avoiding or at least reducing cognitive work – or effort – is called effort discounting [2].’

Comment: This definition of effort discounting is still imprecise (as it is used interchangeably with the ‘law of least mental effort’) and not exactly in line with the reference [2] provided here; see first round of review. It might be best to simply remove this sentence.

AR2.1. We thank the reviewer for bringing up this issue again and for clarifying the case where we have been imprecise. We refrained from removing the sentence because of its importance in the outline of our study, but specified both terms effort discounting and law of less work accordingly. 

„In general, people prefer a less demanding behavioral option when it is rewarded as much as a more demanding one – but not all. Hull [1] coined this principle referred to the general tendency of avoiding or at least reducing energy expenditure or work as the law of less work [1]. Yet, this principle of behavior is not only limited to physical work, but is true for cognitive work as well. The general tendency of avoiding or at least reducing cognitive w Work– or effort – even reduces the value associated with a given payoff – a phenomenon is called effort discounting [2]. While numerous studies …”

RP2.2. p.4, l.68-74: ‘(…) Westbrook et al. [6] experimentally showed that individuals with high need for cognition (i.e., a main indicator of cognitive effort investment) indicated a higher subjective value with demanding tasks than individuals with low need for cognition. Therefore, individuals with high levels in cognitive effort investment would be less likely to increase their effort based on increasing payoff, but rather based on increasing demand. In other words, when given equal demands, individuals with high compared to low levels of cognitive effort investment exert less effort.’

Comment: I appreciate the added section which this paragraph is a part of, with a good explanation of what is meant by efficient effort allocation and evidence presented for the intrinsic motivation of individuals with a high CEI disposition to perform effortful tasks. However, the conclusions in the last part do not strictly follow from the evidence presented. The evidence presented shows that those with high NFC seem to like demanding tasks more than those with low NFC. But this does neither (‘therefore’) indicate that the high-scorers would be less likely to increase their effort based on increasing payoff, nor (‘in other words’) that at equal demands high scorers would exert less effort. At least not directly – more evidence is needed to make this case and these predictions.

AR2.2. We restructured the entire paragraph and moved forward the supporting evidence as foundational evidence. By this, we now present our conclusions more strictly on the entire evidence rather than just the trait concept. This lead to the following revised paragraph:

“Do individuals with high cognitive effort investment actually invest effort more efficiently (i.e., dynamically according to task demands) in task processing and regardless of external incentives (e.g. payoff)? By definition these individuals are highly intrinsically motivated to perform effortful cognition [8], approach and enjoy cognitive challenges while reporting less difficulty and less negative emotions [11] and perform self-controlled behavior relatively effortlessly and unconsciously [12]. For example In line, Westbrook et al. [6] experimentally showed that individuals with high need for cognition (i.e., a main indicator of cognitive effort investment) indicated a higher subjective value with demanding tasks than individuals with low need for cognition. However, individuals with high need for cognition not only value demanding tasks more than those with low need for cognition, but also process these tasks with less effort as demonstrated by Mussel et al. [13]. They found that individuals with high levels of need for cognition, as a main component of cognitive effort investment, allocated their cognitive resources according to task demands, as indicated by frontal midline theta power (FMθ) in the electroencephalogram, while performing better in the harder task. This was not the case for individuals with low levels of need for cognition, who invested a considerable amount of resources even in easy tasks and performed comparably. In a study by Sandra and Otto [14], additional extrinsic motivation in terms of payoff did not add to the already high intrinsic motivation in individuals with high need for cognition to invest effort. They observed that individuals with low levels of need for cognition increased their amount of effort expenditure when facing higher payoff, whereas individuals with high need for cognition did not. Therefore According to the available evidence, individuals with high levels in cognitive effort investment would be less likely to increase their effort based on increasing payoff, but rather based on increasing demand. In other words, when given equal demands payoffs, individuals with high compared to low levels of cognitive effort investment invest effort according to task demands; whereas when given equal demands, they exert less effort, especially under low demand conditions. Thus, their performance would not benefit from payoff in terms of enhanced resource allocation (e.g. stronger increase in frontal midline theta power (FMθ) in the electroencephalogram) in order to improve performance (e.g. faster reaction times). Fig 2 hypothetically illustrates these interactions. 

Fig 2. Expected relation between actual effort investment and (A) demand and (B) payoff as a function of cognitive effort investment.

First evidence that support this notion come from Mussel et al. [13] and Sandra and Otto [14]. The former found that individuals with high levels of need for cognition, as a main component of cognitive effort investment, allocated their cognitive resources according to task demands as indicated by FMθ in the electroencephalogram. This was not the case for individuals with low levels of need for cognition, who invested a considerable amount of resources even in easy tasks. The latter observed that individuals with low levels of need for cognition increased their amount of effort expenditure when facing higher payoff, whereas individuals with high need for cognition did not.”

RP2.3. p.6, l.134-136: ‘In line with our assumption that individuals with high levels of cognitive effort investment find task demands easier compared to individuals with low levels (…)’

Comment: At this point in the paper, this assumption appears a bit surprising as it has not been properly introduced and substantiated with evidence yet. While the authors earlier acknowledge the correlation between e.g. NFC and different cognitive ability/capacity measures, they have thus far focused on justifying why associations between NFC and demand- and payoff-related effort-adjustment are not due to differences in cognitive ability. But they have not clearly stated that they assume task demands will be perceived as easier by those scoring high on CEI.

AR2.3. We reduced the sentence to the main findings of Kramer et al. 

„In line with our assumption that individuals with high levels of cognitive effort investment find task demands easier compared to individuals with low levels, Kramer et al. [20] found a positive relation between need for cognition and performance (d’) for all n-back levels (i.e. 1-back, 2-back, 3-back), .14 < r < .22, in adolescents.“

###### Discussion: ######

RP2.4. p. 44, l. 786-790: ‘In the present study, we were interested in the processes underlying this motivation to seek out effort. We focused on the main questions: (1) Is cognitive effort investment related to objective effort indices? (2) Does cognitive effort investment moderate the relation between effort indices and demand and payoff?’

Comment: Strictly speaking, there is no (direct) relation between the presented hypotheses and the interest in ‘the processes underlying this motivation to seek out effort’. Seeing the other revisions that were made, I am assuming the first sentence here was supposed to be changed as well.

AR2.4. Indeed, we missed to change this sentence and appreciate the reviewer’s careful reading. We changed the sentence accordingly. 

„In the present study, we were interested in the processes underlying this motivation to seek out effort relationship between trait cognitive effort investment and actual effort investment during task processing. We focused on the main questions: (1) Is cognitive effort investment related to objective effort indices? (2) Does cognitive effort investment moderate the relation between effort indices and demand and payoff?“

RP2.5. p.45, l.801-804: ‘Hence, individuals with high levels of effortful self-control generally experience lower effort during task processing, which is consistent with the findings of de Ridder et al. [12] and strengthens our assumptions on individuals scoring high on cognitive effort investment.’

Comment: The last sentence sounds somewhat incomplete – which assumptions exactly does this strengthen?

AR2.5. We now provide more details on the findings of de Ridder et al. and on our assumption. 

 “Hence, individuals with high levels of effortful self-control generally experience lower effort during task processing, which. This is consistent with the meta-analytic findings of de Ridder et al. [12] that self-control correlates more strongly with automatic than controlled behavior, suggesting that individuals with high self-control tend to establish automatisms being unconscious and less effortful. Moreover, and it strengthens our assumptions on individuals scoring high on cognitive effort investment to generally invest effort more efficiently.”

---

## [Decision Letter · Decision Letter 2]

19 Jul 2023

Cognitive effort investment: Does disposition become action?

PONE-D-23-01823R2

Dear Dr. Kührt,

We’re pleased to inform you that your manuscript has been judged scientifically suitable for publication and will be formally accepted for publication once it meets all outstanding technical requirements.

Kind regards,

Árpád Csathó, Ph.D.

Academic Editor

PLOS ONE

Additional Editor Comments (optional):

Reviewers' comments:

Reviewer's Responses to Questions

**Comments to the Author**

1. If the authors have adequately addressed your comments raised in a previous round of review and you feel that this manuscript is now acceptable for publication, you may indicate that here to bypass the “Comments to the Author” section, enter your conflict of interest statement in the “Confidential to Editor” section, and submit your "Accept" recommendation.

Reviewer #2: All comments have been addressed

2. Is the manuscript technically sound, and do the data support the conclusions?

Reviewer #2: (No Response)

3. Has the statistical analysis been performed appropriately and rigorously? 

Reviewer #2: (No Response)

4. Have the authors made all data underlying the findings in their manuscript fully available?

Reviewer #2: (No Response)

5. Is the manuscript presented in an intelligible fashion and written in standard English?

Reviewer #2: (No Response)

6. Review Comments to the Author

Reviewer #2: (No Response)

7. PLOS authors have the option to publish the peer review history of their article (what does this mean?). If published, this will include your full peer review and any attached files.

Reviewer #2: No

---

## [Editor Report · Acceptance letter]

13 Aug 2023

PONE-D-23-01823R2 

Cognitive effort investment: Does disposition become action? 

Dear Dr. Kührt:

I'm pleased to inform you that your manuscript has been deemed suitable for publication in PLOS ONE. Congratulations! Your manuscript is now with our production department. 

Kind regards, 

on behalf of

Dr. Árpád Csathó 

Academic Editor

PLOS ONE